# In-plane reorientation induced single laser pulse magnetization reversal

Y. Peng[1], D. Salomoni [2], G. Malinowski [1] ✉, W. Zhang[1,3,4], J. Hohlfeld[1], L. D. Buda-Prejbeanu [2], J. Gorchon [1], M. Vergès [1], J. X. Lin [1], D. Lacour [1], R. C. Sousa [2], I. L. Prejbeanu[2], S. Mangin [1] & M. Hehn [1] ✉

Single Pulse All Optical Switching represents the ability to reverse the magnetization of a nanostructure using a femtosecond single laser pulse without any applied field. Since the first switching experiments carried out on GdFeCo ferrimagnets, this phenomena has been only recently extended to a few other materials, MnRuGa alloys and Tb/Co multilayers with a very specific range of thickness and composition. Here, we demonstrate that single pulse switching can be obtained for a large range of rare earth–transition metal multilayers, making this phenomenon much more general. Surprisingly, the threshold fluence for switching is observed to be independent of the laser pulse duration. Moreover, at high laser intensities, concentric ring domain structures are induced. These striking features contrast to those observed in Gd based materials pointing towards a different reversal mechanism. Concomitant with the demonstration of an in-plane magnetization reorientation, a precessional reversal mechanism explains all the observed features.

The development of new strategies to perform magnetization reversal at ultra-short timescales fits with the ceaseless demand of future magnetic storage for nonvolatile, energy-efficient, and ultrafast functional memory or logic elements. Historically, the writing of information on magnetic media was performed using a magnetic field. However, this became limiting when reducing the bit size below the hundred nanometers scale and increasing the writing speed above the GHz. Alternative solutions have emerged using spin-polarized currents in nanosized magnetic structures like the spin transfer torque switching[1] or more recently spin–orbit torque switching[2]. Nevertheless, these technologies are limited due to the large increase of the required current density when decreasing the pulse duration, precluding their potential use below 100 ps timescales[3,4].

In 1996, a major discovery of Bigot et al. launched the new research area of femtomagnetism after demonstrating that a femtosecond laser pulse excitation of a thin Ni film leads to a sub-picosecond demagnetization[5]. Almost 10 years later, a complete deterministic all-optical switching (AOS) of the magnetization of ferrimagnetic GdFeCo

alloys was demonstrated using circularly-polarized laser pulses[6]. Subsequently, it was shown in similar GdFeCo layers that a single femtosecond laser pulse could induce a toggle switching independently of the light helicity. The effect named All Optical-Helicity Independent switching (AO-HIS) results from an ultrafast heating process related to distinct dynamics of both the rare earth and transitions metals elements and related to the high transient electron temperature being out of equilibrium with the lattice[7,8]. Nevertheless, the possibility to reverse the magnetization in such alloys using optical or electrical excitations with pulses longer than the electron-phonon relaxation time challenges a mechanism involving the necessity of a strong nonequilibrium electronic state[9–13]. Davies et al. recently pointed out the importance and the competition between the inter-sublattice exchange coupling and the spin-lattice relaxation time[12]. This could potentially explain why AO-HIS has for a long time only been observed in GdFeCo alloys[8], Gd/Co multilayers[14], and GdTbCo[15]. Recently, AO-HIS was reported in half-metallic ferrimagnetic Heusler alloys $Mn_2Ru_xGa$ which possesses two inequivalent Mn sublattices[16]. The

[1]Université de Lorraine, CNRS, IJL, F-54000 Nancy, France. [2]Univ Grenoble Alpes, CEA, CNRS, Grenoble INP, SPINTEC, 38000 Grenoble, France. [3]Anhui High Reliability Chips Engineering Laboratory, Hefei Innovation Research Institute, Beihang University, 230013 Hefei, China. [4]MIIT Key Laboratory of Spintronics, School of Integrated Circuit Science and Engineering, Beihang University, 100191 Beijing, China. ✉e-mail: gregory.malinowski@univ-lorraine.fr; michel.hehn@univ-lorraine.fr

results suggested that the magnetization switching is exchange-driven in agreement with what has been previously reported for the case of GdFeCo[12,17,18].

Surprisingly, Avilés-Felix et al. reported the possibility to reliably toggle switch the magnetization of a Tb/Co multilayer, whose anisotropy is large enough to preserve magnetically stable information bits at small diameters in a magnetic tunnel junction and maintain its perpendicular anisotropy even after a 250 °C annealing process[19,20]. Using a crossed-wedge multilayer structure, they showed that the switching occurs in a very narrow range of Tb and Co thicknesses, that has to be precisely controlled at the Angstrom level, only when a multilayer is used and for pulse durations depending on the Tb and Co thickness ratio. These striking results potentially open the way for their integration into scalable magnetic tunnel junctions but the mechanism leading to single-pulse AO-HIS in these samples remains unclear.

In this paper, we present a detailed and extensive study regarding single-pulse AOS of magnetization in a large variety of samples such as magnetic multilayers and alloys containing different rare earth (RE) (Tb, Dy) and transition metals (TM) (Co, Py, and Fe) elements. We demonstrate that the toggle switching obtained in Co/Tb is robust for laser pulses as long as 12 ps and with respect to layer thickness (up to few tens of nanometers) or TM/RE chemical nature when Tb is replaced by ferrimagnetic alloys. Furthermore, we show that the magnetization reversal process is completely different from the one observed in GdFeCo or Mn$_2$Ru$_x$Ga alloys and involves an in-plane magnetization reorientation and potentially a precessional mechanism.

## Results

As a starting point in our study, we consider [Tb/Co]$_5$ and [Tb/Fe]$_4$ magnetic multilayers, similar to the one used by Avilés-Felix et al.[19,20] (5 and 4 is the number of repetitions). These multilayers exhibit a strong perpendicular magnetic anisotropy at room temperature and the magnetization of the Tb and Fe or Co layers are antiferromagnetically exchange-coupled at the interface. Typical results obtained for a [Tb(1.31 nm)/Fe(1.89 nm)]$_4$ multilayer are presented in Fig. 1 (the study as a function of the thickness of Tb and Fe is reported in Supplementary Fig. 1). Tb/Co multilayers present the same behavior as reported in Supplementary Fig. 2a (the study as a function of thickness is shown in Supplementary Fig. 2b). As depicted in Fig. 1a, toggle switching of a single magnetic domain is observed for excitation with a pulse duration of 50 fs and a low fluence of 1.9 mJ/cm². More interestingly, for laser fluence larger than 2.6 mJ/cm², a bullseye structure starts

appearing with opposite magnetic directions in adjacent rings. Increasing the laser fluence results in a higher number of rings. As shown in Fig. 1b, the central domain is alternatively being reversed, and this up to the higher fluence (7.0 mJ/cm²) for which a multidomain state is stabilized. This striking behavior has been studied systematically as a function of the laser pulse duration, which allows us to establish a complete state diagram (pulse duration versus fluence) reported in Fig. 1c. This state diagram resembles neither the state diagram observed in the case of AO-HIS in GdFeCo[13] nor the state diagram observed in the case of multiple pulse All Optical Helicity Dependent Switching (AO-HDS)[21]. Surprisingly, the fluences required to reverse and stabilize a given number of rings depend very little or not at all on the duration of the laser pulses. Single laser pulse-induced magnetization switching is observed for pulse durations up to 12 ps, the longest pulse duration reachable in our ultrafast laser equipment. Both Tb/Co and Tb/Fe multilayers behave similarly, both having ring structures and low dependence of the reversal fluences on the laser pulse duration. Both features will be considered to determine the conditions to obtain the reversal and to discuss the possible mechanism at its origin.

The ring structure does not seem to have a dipolar origin i.e., linked to the magnetostatic coupling between adjacent magnetic domains. As reported in Fig. 2, we performed two successive pulses by slightly moving the spot: the local magnetic state is found to be independent of the configuration of the neighboring first domains and depends only on the distance to the spot center. For the fluence used in the experiment shown in Fig. 2, region 2 experiences 2 reversals, one for each pulse, region 1 (respectively 1′) experiences reversal for pulse 1 (respectively 2) and the state of region 0 does not change. Previously, similar ring structures have been observed in ferrites, and they have been attributed to the reorientation of the magnetization from out of plane to in-plane followed by its precession[22,23]. In metallic thin films[24], magnetization precession could be observed without magnetization reversal. In both cases, for ferrites and metallic thin films, an in-plane magnetic field was applied which is not the case in our study.

The fact that the critical fluences (TF$_1$, TF$_2$,...) are independent of pulse duration, implies that the underlying mechanism cannot be governed by the maximum electronic temperature which decreases with increasing the pulse duration for a given fluence, but by the energy transferred to the lattice. Ultimately, what matters in the reversal mechanism is the total amount of energy pumped into the system, independently of its rate. Therefore, the energy density after equilibration of electrons-spins-phonons is likely to govern the

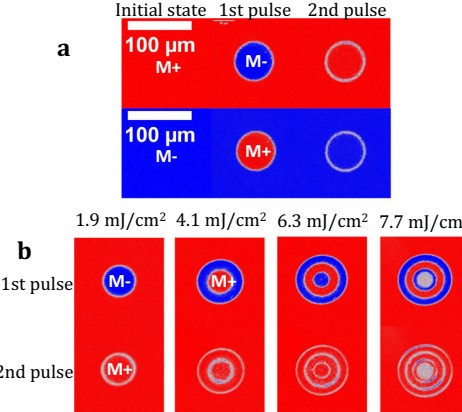

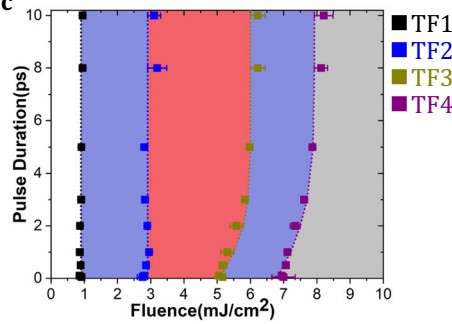

**Fig. 1 | Single-pulse switching and state diagram for [Tb(1.31 nm)/Fe(1.89 nm)]$_4$ multilayer under zero applied field.** M+ (respectively, M−) corresponds to magnetization pointing perpendicular to film plane, along +z (respectively −z) in red (respectively in blue). **a** Background subtracted images after each single pulse with 1.9 mJ/cm² laser pulse at 50 fs, starting from M+ or M- state. **b** Background subtracted images after first single with 50 fs laser pulse of different fluence. **c** State diagram of reversal pulse duration versus fluence. TF$_1$ is the border for switching; TF$_2$ is the border to two domains state (i.e., one ring); TF$_3$ is the border to three domains state (i.e., two rings); TF$_4$ is the border to multidomain state at the center (i.e., three rings). The dotted lines are guide for the eyes.

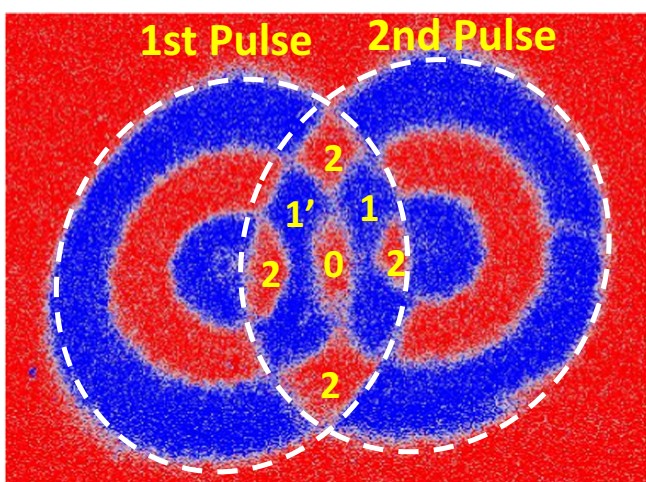

**Fig. 2 | Domain pattern obtained by sending two successive pulses: a first one and then, after moving the laser at a distance smaller than the spot size, a second one.** The two spots are then overlapping. The experiments were done on a [Tb(1.31 nm)]/Fe(1.89 nm)]$_4$ multilayer. The color scale corresponds to magnetization pointing out of plane: M+ (respectively M−) along the perpendicular to the film plane, along +z (resp. −z) in red (resp. in blue). The sample is initially saturated along the M+ direction. The laser pulse has a fluence of 5.2 mJ/cm² and a duration of 50 fs.

reversal. To go deeper into this intuitive description, we performed two-temperature (2T) model simulations[25] (Supplementary Methods). We did not include the spin bath which typically follows closely the temperature of electrons, whereas the lattice is generally much slower. As expected, increasing the fluence rises both electron, $T_e$, and phonon, $T_{ph}$, temperatures but, with constant fluence, the $T_e$ peak reduces drastically upon increasing the pulse duration while $T_{ph}$ remains constant. Increasing $T_{ph}$ above a threshold could lead to a modification of the magnetic anisotropy and induce the magnetization precessional switching for which a full demagnetization of the TM is not mandatory. The transfer of energy to the phonon is here justified by the strong spin–orbit coupling characteristic for Tb or Dy, contrary to Gd.

In the following, we will test the hypothesis of a spin reorientation accompanied by a precession, highlighting the elements required to obtain a reversal in a single pulse and optimize the material parameters. In the case of the Tb/Co and Tb/Fe multilayers, Tb is magnetic at room temperature only because of its proximity or contact with the Co or Fe layers[26]. Thus, modifying its thickness will also lead to a variation of its anisotropy and its magnetic moment. In order to check if thicknesses or the variation of the magnetic properties are key parameters in the process, as reported in refs. 19,20, the Tb layer was replaced by a rare earth–transition metal (RE–TM) alloy with a high RE concentration so the net magnetization of the alloy points into the direction of the magnetization of the RE sublattice. Due to the high RE content, the alloy has a Curie temperature higher but close to room temperature[27] and possess a perpendicular to film plane magnetic anisotropy (PMA). We will compare the measurement results obtained on [Tb/Co], the multilayer in which this reversal has been discovered[19,20], and on [Tb/Fe] multilayer reported here to [TbCo/Co], [TbCo/Ni$_{80}$Fe$_{20}$], [TbCo/Co$_{40}$Fe$_{40}$B$_{20}$], [DyCo/Co], [TbFe/Co] multilayers. In the following, Ni$_{80}$Fe$_{20}$ (respectively Co$_{40}$Fe$_{40}$B$_{20}$) is referred as Py (respectively CoFeB). The complete study indicates that all these combinations of multilayers are showing a similar all-optical switching behavior, i.e., a single-pulse magnetization reversal with ring structure and a magnetization reorientation in-plane compatible with a precessional process for the reversal. The list of all the samples tested is given in Table 1.

Finally, if the in-plane reorientation of the magnetization of the ferromagnetic layer (Py, CoFeB, or Co) is a key ingredient to promote the toggle reversal of the magnetization of the whole multilayer stack, PMA anisotropy in the FM layer has to vanish during the process. This is not the case when Co, CoFeB, or Py layers are in contact with a Pt buffer or capping layer or with an insulating MgO layer in which case anisotropy is high. Therefore a multilayer structure appears necessary for the proposed switching mechanism. Since PMA in TbCo, DyCo or TbFe alloys is expected to disappear as the lattice temperature rise after the laser pulse, the simplest multilayers to characterize are then TbCo/X/TbCo (X = Co, Py, CoFeB), Y/Co/Y (Y = DyCo, TbFe) trilayers in which Co, Py, and CoFeB PMA is induced by its proximity with TbCo, DyCo, or TbFe alloys.

The results obtained on the Tb$_{32}$Co$_{68}$/Co(1.52 nm)/Tb$_{32}$Co$_{68}$ trilayer are reported in Fig. 3. In addition, the results obtained on the Tb$_{32}$Co$_{68}$/Co(wedge)/Tb$_{32}$Co$_{68}$, Dy$_{35}$Co$_{65}$/Co(wedge)/Dy$_{35}$Co$_{65}$, Tb$_{55}$Fe$_{45}$/Co(wedge)/Tb$_{55}$Fe$_{45}$, Tb$_{32}$Co$_{68}$/Py(wedge)/Tb$_{32}$Co$_{68}$ and Tb$_{32}$Co$_{68}$/CoFeB(wedge)/Tb$_{32}$Co$_{68}$ trilayers are given in Supplementary Fig. 4a–e, respectively. In all the trilayers studied, the thickness of the ferrimagnetic alloy was fixed to 4 nm. In Tb$_{32}$Co$_{68}$/Co(wedge)/Tb$_{32}$Co$_{68}$, single-pulse reversal can occur for Co thickness above 1.81 nm. The reversal is repeatable for up to 30,000 consecutive laser pulses and the characteristic ring structure is always present. For a Co thickness less than 1.5 nm, only a multidomain state has been observed. The same trend is seen for TbCo, DyCo, and TbFe-based trilayers, where a minimum FM thickness is required to obtain the reversal (see Supplementary Fig. 4). Our findings agree with results reported in the literature, for TbCo or DyCo single-layer alloy[28], which does not demonstrate single-pulse switching. However, the presence of an additional exchange-coupled FM layer makes single-pulse optical switching possible. Very importantly, our findings demonstrate undoubtedly that we can largely extend the domain of thicknesses in which the single laser pulse reversal exists. In Tb$_{32}$Co$_{68}$ (4 nm)/Co($t$ nm)/Tb$_{32}$Co$_{68}$ (4 nm), single-pulse reversal appears for t ranging from 1.5 to 2.3 nm. With the use of CoFeB (respectively, Py), this thickness can be extended up to 2.87 nm (respectively, 5.6 nm). This is well beyond the limited range of thicknesses for which the AOS switching was observed for Tb/Co multilayers.

One common feature of all trilayers and multilayers, which exhibit the single laser pulse reversal, is the reorientation of the magnetization in the plane. In Fig. 4, we report the magnetization dynamics for three different systems, the historical Tb/Co multilayer as well as the TbCo/Co/TbCo trilayers and [TbCo/Co] and [DyCo/Co] multilayers. This dynamic behavior is obtained by measuring the Time-Resolved Magneto Optic Kerr images (TR-MOKE). The polar geometry used in this measurement allows mainly to probe the perpendicular to film plane component of transition metal magnetization. The stroboscopic measurement requires resetting the magnetic configuration after each pulse. Therefore, a continuous (DC) magnetic field is applied during the measurement and, therefore the precession occurs around the direction of the effective magnetic field equal to the sum of the anisotropy field and the external field. The measurements show that after a fast demagnetization at a time scale of some 100 fs, there is first a recovery of the magnetization, similar to the conventional demagnetization/re-magnetization shown in various ferromagnetic and ferrimagnetic materials. At longer timescales, a few tens of picoseconds, a magnetization decrease is observed. This decrease cannot be attributed to the creation of a domain structure because the Kerr microscope images do not reveal any domain formation and the experiment is done under an applied field which prevails over the possibility of a non-uniform magnetic configuration (see Supplementary Fig. 5). This decrease is attributed to the reorientation of the magnetization in the plane of the layers. To further reinforce this statement, pump-probe dynamical hysteresis loops are reported in Supplementary Fig. 12 for a sample with the composition TbCo(4)/Co(2)/TbCo(4). It is clearly

**Table 1 | Series of stacks that have been studied in this paper: 1–11 are multilayers, 12–14 are bilayers, and 15–19 are trilayers**

| | Stacks | | | | | | | | |
|---|---|---|---|---|---|---|---|---|---|
| 1 | [Tb(wedge)/Fe(wedge)]*4 | | | | | | | | |
| 2 | [Tb(wedge)/Co(wedge)]*5 (Cross wedge) | | | | | | | | |
| 3 | [$Tb_{40}Co_{60}(x)$/Co(wedge)]*3 | x = 1 | | | | | | | |
| 4 | | x = 2 | | | | | | | |
| 5 | | x = 3 | | | | | | | |
| 6 | | x = 4 | | | | | | | |
| 7 | | x = 5 | | | | | | | |
| 8 | | x = 7.5 | | | | | | | |
| 9 | [$Dy_yCo_{100-y}$ (4) /Co(wedge)]*3 | y = 25 | | | | | | | |
| 10 | | y = 30 | | | | | | | |
| 11 | | y = 35 | | | | | | | |
| 12 | Pt(5) /Co(wedge)/ $Tb_{32}Co_{68}$ (4) | | | | | | | | |
| 13 | Pt(5) /Co(wedge)/ $Tb_{32}Co_{68}$ (8) | | | | | | | | |
| 14 | Pt(5) /Co(wedge)/ $Tb_{32}Co_{68}$ (16) | | | | | | | | |
| 15 | $Dy_{35}Co_{65}$ (4) /Co(wedge)/ $Dy_{35}Co_{65}$ (4) | | | | | | | | |
| 16 | $Tb_{32}Co_{68}$ (4) /Co(wedge)/ $Tb_{32}Co_{68}$ (4) | | | | | | | | |
| 17 | $Tb_{32}Co_{68}$ (4) /CoFeB(wedge)/ $Tb_{32}Co_{68}$ (4) | | | | | | | | |
| 18 | $Tb_{55}Fe_{45}$ (4) /Co(wedge)/ $Tb_{55}Fe_{45}$ (4) | | | | | | | | |
| | **Thickness of the TM layer** | | 0.8 nm | 1.2 nm | 1.6nm | 2.0 nm | 2.4 nm | 2.8 nm | 3.2 nm |
| 19 | $Tb_{32}Co_{68}$ (4) /Permalloy (wedge)/ $Tb_{32}Co_{68}$ (4) | | | | | | | | |
| | | | 2.0 nm | 2.6 nm | 3.2nm | 3.8 nm | 4.4 nm | 5.0 nm | 5.6 nm |

Light pink color shows the TM layer (including Co, Fe, CoFeB, Permalloy) thickness range that have been studied in this paper, while blue color shows the region of TM thickness where single-shot switching occurs in different stacks. Note that stack 1 is a double wedge sample in the same direction, while stack 2 is a cross-wedge sample. Numbers shown within the brackets are the layer thicknesses in nanometers.

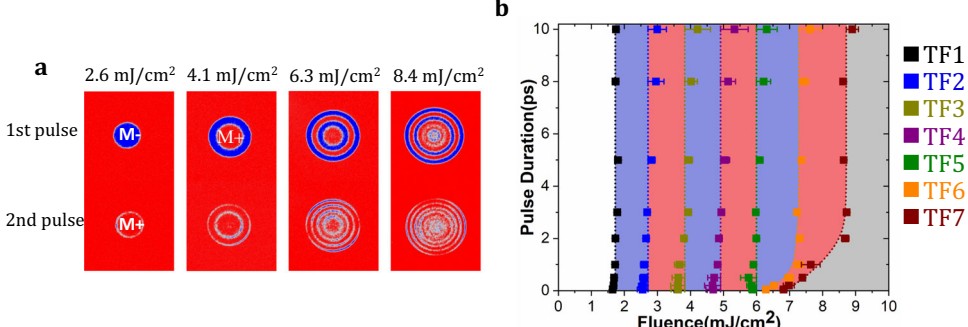

**Fig. 3 | Single-pulse reversal in $Tb_{32}Co_{68}$(4 nm)/Co(1.52 nm)/$Tb_{32}Co_{68}$(4 nm) trilayer. a** Background subtracted images after first and second pulses with 50 fs laser pulse with different fluences. **b** State diagram which is plotted by pulse duration as a function of fluences. $TF_1$ is the border for switching; $TF_2$ is the border to two domains state, one ring; $TF_3$ is the border to three domains state, two rings; etc....$TF_7$ is the border to the demagnetized state at the center. The dotted lines are guide for the eyes.

observed that the hysteresis loop measured in the polar geometry is characteristic of an out-of-plane magnetization without any pump. Instead the hysteresis loop measured at 10 or 30 ps are characteristic of an in-plane magnetization. So, undoubtedly, the magnetization reorients in the plane. This reorientation seems to be a common feature and supports the hypothesis of the possible precession of the magnetization. The existence of the DC field, in combination to a strongly damped reorientation of the anisotropy, makes the

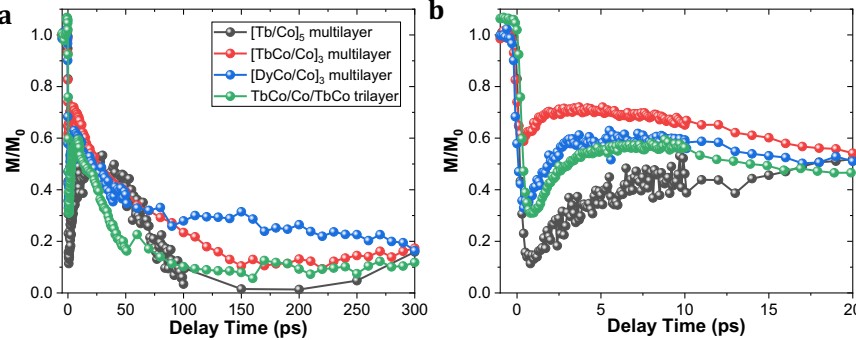

**Fig. 4 | TR-MOKE measurements performed on the different multilayers showing single-pulse reversal. a** 300 ps time scale. **b** Zoom-in of the first 20 ps time scale. The description of the stacks and conditions for TR-MOKE measurements have been shown in Supplementary Table 1.

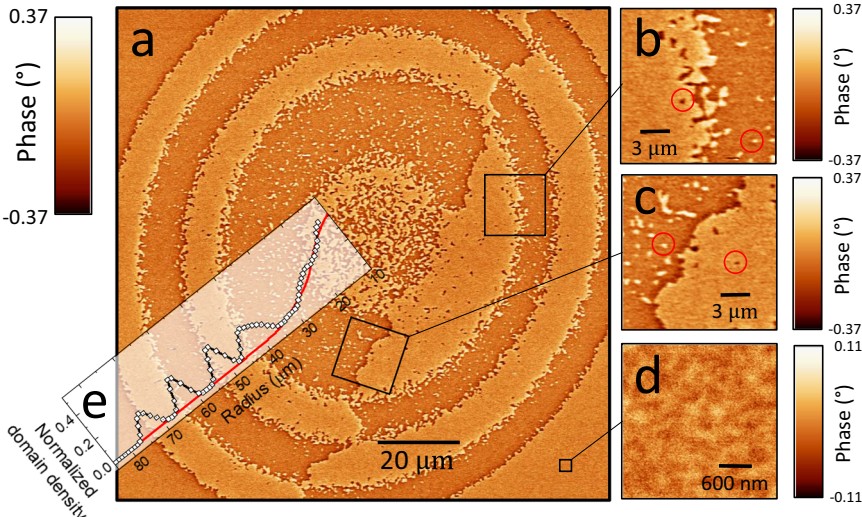

**Fig. 5 | Magnetic force microscopy investigation of a characteristic ring structure. a** A ring structure observed by MFM. **b** Zoom on the region that separates two rings. **c** Zoom on a domain wall. **d** Zoom on a region outside the laser spot that has been saturated before the experiment. **e** Annular analysis of the small domain density.

precession difficult to be observed. However, in some cases, they could be observed as show in Supplementary Fig. 10 supporting again the precession mechanism.

Regarding a magnetization reorientation, suppressing the source of anisotropy of the Co or weakening its contribution is a key ingredient for robust AOS. This is achieved in the trilayer in which the Co anisotropy is given by the 4 nm thick TM-RE alloy at both interfaces. In a MTJ, the FM layer in contact with the tunnel barrier has a strong anisotropy, so to get the reorientation either a multilayer is needed, or the thickness of the TM-RE has to be increased. We tested these two strategies and the results are reported in Supplementary Figs. 6–9. We clearly show on the one hand that increasing the number of bilayers in the multilayer, and thus the number of Co layer in contact with an alloy layer, increases the quality of the reversal. On the other hand in a single TbCo/Co bilayer, the switching is favored when the thickness of TbCo is large compared to the Co layer. This allows obtaining a reversal in a multilayer with a record thickness of 40 nm.

While the in-plane reorientation appears as a key condition for reversal, it is not sufficient to explain alone a precession-type reversal. An effective field around which the precession takes place has to be identified[22,23]. One source of symmetry breaking that we can see in the system is the laser's Gaussian profile. Although the laser profile has a cylindrical symmetry, exposure is not homogeneous. Therefore, lateral gradients of deposited energy are created in the film. In the static domain structure, we analyze the position of the domain boundaries

that separate the rings. We can observe that the walls are always located at the same local threshold energy and do not depend on their distance to the Gaussian beam's center (see Supplementary Fig. 11). The energy gradients do not seem to dominate the switching mechanism, the locally absorbed fluence does. We can therefore neglect dipolar interactions or lateral-induced strain as a possible source for this in-plane effective field. However, this observation is again compatible with a precessional reversal mode and a similar conclusion has been proposed in Co/Tb multilayers, in a very narrow range of Tb and Co thicknesses, during the reviewing process of our paper[29].

A close analysis of a Magnetic Force Microscopy images of the rings domain structure helps to get a view of the process (Fig. 5a). The reversal process has to be considered at the scale of a grain or couple of grains in the TM layer. First, we can clearly observe inside the rings the presence of small domains, with sizes ranging from 100 to 300 nm (Fig. 5b, c, red circles). Their density increases upon increasing the fluence i.e., toward the center of the spot (Fig. 5e). Second, the rings are not separated by sharp and well-defined domain walls (Fig. 5b). Instead, the separation is made of a collection of small domains whose density is the highest at the transition (Fig. 5e). In comparison, a domain wall located at the center of Fig. 5 (and exiting in the sample prior to laser irradiation) is much sharper (Fig. 5c).

Most of the samples studied (numbers 3–19 in Table 1) are made of a (TM-RE alloys/FM) multilayer. From the literature, it is known that

TM-RE alloys are amorphous, while FM layers are polycrystalline or textured when deposited at room temperature on amorphous layers, with typical grain sizes of 20–30 nm. While the TM-RE alloys have PMA, we expect a local distribution of the direction of anisotropy linked to the polycrystalline nature of the FM layers. The consequences of this distribution are visible in the MFM image recorded in the area of the layer which has been saturated before the experiment (Fig. 5d). The fast reduction and then recovery of the anisotropy with the temperature in the TM-RE alloy, related to the very low Tc of the RE, leads to the orientation of the magnetization of the FM layer in-plane and to its precession. The key ingredient for the reversal is the persistence of the magnetization of the FM layer due to its high Tc (since it is made of a pure Co, Fe, Py, CoFeB layer). In a scenario of magnetic grain or couple of grains (identified as a "magnetic" grain) behaving independently, the magnetization will precess around the local anisotropy axis which is distributed from grain to grain.

Starting from close to the perpendicular axis, the precession would occur around this local anisotropy and when the equatorial plane of the film is crossed, reversal occurs. The time spent to precess is directly linked to the energy brought by the laser beam-defining regions, always located at the same energy (Supplementary Fig. 11), in which the magnetization of grains can either stop its precession along the starting or reversed orientation. This defines the position of the rings limits and also the areas where the small domain density is the highest (Fig. 5). Sometimes, inside the rings, small domains are not reversed and are certainly at positions for which the magnetic grains do not have the suitable anisotropy configuration. Going closer to the center, the density of small domains in the ring increases. At higher fluences, the reversal involves multiple switching which are more sensitive to any distribution of properties as the effective anisotropy axis.

In conclusion, single-pulse all-optical reversal has been extended to many different materials with laser pulse duration up to 10 ps. It is a relatively ubiquitous phenomenon in multilayers with optimized design. The picture of precessional reversal emerges due to the strong evidence that magnetization has to reorient in-plane before magnetization reverses. The resulting state diagram resembles neither the state diagram observed in the case of AO-HIS in GdFeCo nor the state diagram observed in the case of multiple pulse AO-HDS and demonstrates the existence of a new type of AOS. In terms of reversal time, this single-pulse precessional like AOS possesses typical timescales between those of GdFeCo and multiple pulse AO-HDS. While the in-plane reorientation of the magnetization is undoubtedly proven and is one key ingredient, a precession at the scale of a grain or couple of grains in the TM, which keeps a sizable magnetization thanks to its high Tc, leads to the reversal in a ring shape domain structure. This is the second key ingredient needed, that is missing if only the RE–TM alloy is used.

## Methods

### Deposition conditions and wedge fabrication

The multilayers were produced by sputtering in an AJA machine. The alloys are made by co-deposition of different sources arranged in a confocal geometry, the composition being adjusted by the powers applied on the different cathodes. The homogeneity of the layer is obtained by rotating the sample.

Thickness wedges are obtained by stopping the rotation, resulting in a gradient of thickness. A calibration of the thicknesses as a function of the position has been done beforehand, which allows to study the magnetic properties as a function of the thickness in a controlled way and on a single sample. The calibration has been done using a thick Cu layer (see Fig. 6). The wedge-shaped Cu sample spanned of 50 mm. At position 25 mm, the thickness was targeted to be 200 nm. The thickness of Cu has been measured using atomic force microscopy. The variation of the thickness of Cu was fitted as $t_{Cu} = 110.9 + 3.83x$, where

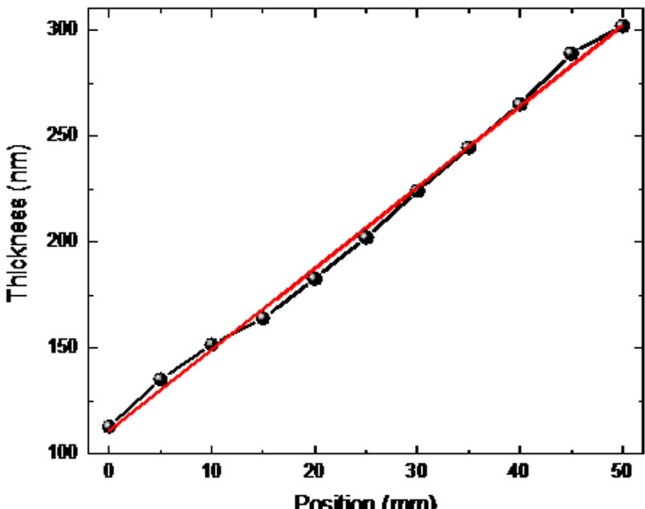

**Fig. 6 | Calibration of the wedge deposition thickness.** Variation of the thickness of a thick Cu wedge as a function of position.

$x$ is the position in mm, as shown in figure below. Thus, the thickness $t$ in the wedge can be obtained by $t(x) = \frac{110.9}{200} \times t_{25\,mm} + t_{25\,mm} \times \frac{3.83}{200}x$, where $t_{25\,mm}$ is the expected thickness at 25 mm. The uncertainty of the thickness is thus $\Delta t(x) = t_{25\,mm} \times \frac{3.83}{200}\Delta x$ where $\Delta x$ is the uncertainty in the position in the sample. $\Delta x$ was estimated to be 0.1 mm. Therefore, $\Delta t(x) \approx 0.002\, t_{25\,mm}$. Since wedges have typical $t_{25\,mm} < 5\,nm$, $\Delta t(x) < 0.01$. That is the reason why the layer thicknesses are quoted to hundredths of nanometers.

### Laser setup and fluence measurements

**AOS measurement.** Ti:sapphire femtosecond laser source and regenerative amplifier were used for the pump laser beam in AOS measurement. The wavelength and repetition rate of the femtosecond laser are 800 nm and 5 kHz, respectively. LED light probe source with a wavelength of 628 nm was used for taking MOKE images. The Gaussian beam size was determined in two ways: by directly observing the beam at the focal plane of the microscope's lens (Used for imaging) and by using the domain size vs pulse energy fit (see "Data analysis"). For samples grown on the glass substrate, pump laser excitation was done on one side of the sample and MOKE microspore observation was on the other side (transmission configuration). Both methods are possible to determine the beam size and provide consistent results. For samples grown on the silicon substrate, the pump and probe light are both on the sample side (reflection configuration). In this case, the beam size was only obtained by fitting the domain size vs pulse energy.

**Data analysis.** The measured laser incident fluence was calculated using

$$\bar{F} = \frac{P}{fS}$$

where $P$ is the measured power, $f$ is the repetition rate of the laser ($f = 5$ kHz), and $S$ is the beam spot area.

For transmission configuration, the beam spot area is

$$S = \pi R^2$$

where $R$ is the radius of the beam spot and equals to $\frac{FWHM}{2\sqrt{(\ln 2)}}$, where FWHM is half of the laser spatial full-width half-maximum. So, the measured fluence $\bar{F}$ is defined such that the power of the laser is

divided by a factor $e^2$ at a distance from the center of the beam:

$$\bar{F} = \frac{P}{f\pi R^2}$$

The fluence profile has then the form:

$$F(r) = 2\bar{F}e^{-\left(\frac{r}{R}\right)^2}$$

where $r$ is the domain radius. The peak value is then twice its measured value. A magnetic domain starts to appear as soon as the fluence peak value exceeds a certain threshold value $F_{th}$. When this happens, one would measure a fluence $\bar{F} = \frac{F_{th}}{2}$. The domain radius must verify the following equation:

$$F(r) = F_{th} \leftrightarrow r = R\sqrt{\frac{1}{2}\ln\left(\frac{\bar{F}}{\bar{F}_{th}}\right)}$$

The threshold powers, as well as beam spot radius, can be extracted as a fitting parameter:

$$r = R\sqrt{\frac{1}{2}\ln\left(\frac{P}{P_{th}}\right)}$$

For the reflection configuration, the domains as well as the beam spot are elongated due to an angle of incidence of 45°. We used the following equation to extract the threshold powers and beam spot area as a fitting parameter: Domain area $= S\ln(\frac{P}{P_{th}})$

### Modeling of electron and phonon temperature to explain the specific state diagram

The specific features of the shape of the experimental state diagrams can be understood in the frame of a two-temperature (2T) model[25]. If we assume that all-optical switching is due to a rapid increase of temperature above a certain threshold, we can draw some interesting points.

The standard 2T model uses the electron and phonon bath to describe the heat components in the system. Each bath has its own heat capacity $C_i(i = e, ph)$ and exchanges energy through the coupling parameter $G$. $P(t)$ describes the laser pulse excitation that heats the electron bath and the two coupled equations read as:

$$C_e \frac{dT_e(t)}{dt} = -G\left[T_{ph}(t) - T_e(t)\right] + P(t) - \frac{C_e}{\tau_0}\left[T_e(t) - T_0\right]$$

$$C_p \frac{dT_{ph}(t)}{dt} = -G\left[T_e(t) - T_{ph}(t)\right] - \frac{C_{ph}}{\tau_0}\left[T_{ph}(t) - T_0\right]$$

$$P(t) = \frac{F}{t_{FM}\tau_L} \cdot e^{\left[-\frac{(t-t_0)^2}{\tau_L^2/(4\ln2)}\right]}$$

Here, $\tau_L$ is the laser pulse duration, $F$ is its fluence, $t_0$ is the moment of the laser pulse application, $t_{FM}$ is the thickness of the magnetic film, $\tau_0$ is the cooling characteristic time pulse to recover the initial temperature of the system $T_0$ .

Supplementary Fig. S3 shows the time evolution of the electron temperature $T_e$ and phonon temperature $T_{ph}$, respectively, upon varying the laser fluence and/or pulse duration. As expected, increasing fluence rises both electron and phonon temperatures but, interestingly, with constant fluence the electron temperature peak reduces drastically upon increasing the pulse duration (e.g., from 100 fs to 1 ps at $F = 5.0$ mJ/cm$^2$ the $T_e$ peak drops of ~500 K).

The AO-HIS mechanism in GdFeCo alloy has been related to a difference in the characteristic demagnetization time of the two sub-lattices (Gd and Fe/Co) that have to demagnetize almost completely in order to switch. Because the TM sublattice has the higher $T_C$ and its magnetic properties come from the 3d electrons, it makes sense that, to switch, the electron temperature $T_e$ needs to approach $T_C$ and thus due to dependency on the pulse duration shown in Fig. 1, the resulting state diagram has a triangular shape[13]. In contrast, for a reorientation of magnetization and precessional switching it is not required to completely demagnetize the TM, or at least it is not crucial, but what seems to be necessary is that the phonon temperature $T_{ph}$ reaches a certain limit value. The dependency on the phonon temperature instead of the electron temperature $T_e$ is justified by the strong spin–orbit coupling of the Tb, contrary from Gd. Supplementary Fig. S3c shows the threshold fluence needed to reach a limit value either for the electron temperature $T_e$ or the phonon temperature $T_{ph}$. The difference between the two trends is similar with our experimental state diagrams in [Tb/Co] and GdFeCo[13].

## Data availability

The data is available from the corresponding author on request.

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

## Acknowledgements

We thank Bert Koopmans, Jeff Bokor, Martin Weinelt, Nicolas Bergeard, and Eric E. Fullerton for fruitful discussions. We also acknowledge financial support from the ANR (ANR-17-CE24-0007 UFO project), the Region Grand Est through its FRCR call (NanoTeraHertz and RaNGE projects), by the impact project LUE-N4S part of the French PIA project "Lorraine Université d'Excellence", reference ANR-15IDEX-04-LUE and by the "FEDER-FSE Lorraine et Massif Vosges 2014-2020", a European Union Program. D.S. has received funding from the European Union's Horizon 2020 research and innovation programme under Marie Skłodowska-Curie grant agreement No. 861300 (COMRAD). W.Z. gratefully acknowledge the National Natural Science Foundation of China (Grants No. 12104030), China Postdoctoral Science Foundation (Grants No.2022M710320) and China Scholarship Council for their financial support of this work.

## Author contributions

M.H., S.M., L.P., and R.S. conceived the study. M.H. and D.S. fabricated the samples and M.H. optimized them. Y.P. performed magnetic measurements. Y.P., D.S., J.X., and W.Z. the fast optics measurements with the help of G.M., J.H., M.V. and J.G. L.D.B.-P. and D.S. performed simulations. M.H. and G.M. wrote the manuscript with input from all authors. All the authors participated to the scientific discussions.

## Competing interests

The authors declare no competing interests.
