## [Peer Review File · Nature Communications]

Reviewers' Comments:

Reviewer #1:

Remarks to the Author:

Review report for "In plane reorientation induced single laser pulse magnetization reversal in rare-earth based multilayer"

The authors of Y. Peng et al. reported all-optical switching in Tb-based multilayers and claimed a new switching mechanism of in-plane reorientation. To argue the in-plane reorientation, they presented three major pieces of evidence: 1) switching is nearly independent on the pulse duration; 2) the switched domain shows concentric ring structures; 3) time-resolved dynamics show a gradual decreasing behavior. However, I am not convinced that these evidences undoubtedly support their claim.

1. pulse duration dependence

The authors showed that optical switching is nearly independent on the pulse duration. They claimed that the switching mechanism is not driven by the nonequilibrium electronic state and the state diagram of Fig. 1c is different from that of the optical switching in Gd-based system. I agree that the nonequilibrium state might be not a critical factor, but I am not convinced that this result can support the new switching mechanism. For example, J. Gorchon et al. reported an optical switching in GdFeCo alloy varying a pulse duration and showed that the critical fluence increases slightly with a wide pulse duration of 55 fs \sim 15 ps [Fig. 4 of PRB 94, 1844006 (2016)]. Therefore, in my opinion, a similar switching diagram (Fig. 1c) has been observed in a conventional Gd-based system, in which the switching mechanism is known to be related to a difference in the demagnetization time of the two sub-lattices.

2. Concentric ring structure

The concentric ring structure of the switched domains is indeed different from that of Gd-based system, but it looks similar to that of Iron garnet system. C. S. Davies et al. reported the concentric ring structure after the optical switching in iron garnet [PRL 122, 027202 (2019)]. This ring structure was explained by nonlinear magnetization precession, which is controlled by laser heating and external magnetic field. According to Davies et al., to have the magnetization precession, an external in-plane field should exist. However, the optical switching of this manuscript was done without an external field. The authors of this manuscript tried to find the physical origin that acts as an effective in-plane field but could not identify it. Therefore, except that the ring structure is similar to that of iron garnet, the authors could not provide a plausible explanation for how the ring structure is driven by in-plane reorientation.

3. Time-resolved dynamics

The authors showed an unexpected decrease in magnetization after a few ps and claimed that it is due to the in-plane reorientation. Although this decrease cannot be explained by simple temperature analysis, I am not convinced that it comes from the in-plane reorientation. When the laser heating induces a sudden change in magnetic anisotropy and causes an in-plane reorientation, the magnetization dynamics show multiple precessions [PRB 97, 014422 (2018)]. However, Fig. 4 of this manuscript does not show any precessional motion. Why? In addition, time-resolved dynamics were done with an external field, whereas optical switching was done without a field. With such a different condition, how one can be sure that the same mechanism governs both experiments?

In summary, the novelty and clarity of this work are not high enough to be published in a high-impact journal of Nature Communications. Therefore, I recommend authors collect more clear evidence to support their claim.

Reviewer #2:

Remarks to the Author:

In general, it is a well written paper, placed well in relation to recent works, that brings interesting new findings to the field. The explanation of the results is somewhat weak, in that the authors propose a few mechanisms but fail to conclusively support one. In addition, some of the supporting data is scattered, and seems to vary too many parameters to have any kind of systematic significance. However, I do believe that the main findings are fundamentally new and

would be of great interest to the community. Therefore, I would like the authors to address the following concerns and possibly re-organize the manuscript to make it more readable before recommending the article for publication in Nature Communications.

The paper shows very systematically that all optical switching with a single laser pulse with up to 10 ps of duration is a relatively ubiquitous phenomena and that it seems to depend on certain material properties of the materials involved. It would be helpful to state this more clearly in the introduction/conclusion.

One very interesting finding of the paper is that the flipping of the magnetic order is not dependent on pulse duration (up to 10ps) or peak energy, but rather the total amount energy deposited in the system. The authors follow up on this with a 2T model, suggesting that it is the lattice temperature, rather than the electron temperature that is the determining factor in the magnetic reversal process. It is unclear why the authors did not attempt to use a 3 temperature model, since excitation of the spins could also be a relevant timescale. The authors should at least justify the fact that they did not use a 3T model here.

The authors state that the importance of the transfer of energy to the phonon is justified by the strong spin orbit coupling present for Tb or Dy, contrary to Gd. However, they also state that a much more robust toggle switching is obtained when Tb is replaced by ferrimagnetic alloys with thicknesses up to few tens of nanometers and for laser pulses as long as 12 ps. They do not offer an explanation for why this is, and the spin-orbit coupling explanation would clearly not work here, since typically these alloys would have a weaker spin-orbit coupling.

In Figure 4 and Table 2 the authors plot the time dynamics for several different compounds. However, the experimental conditions were not the same for the different compounds, which is why they have to be given in Table 2. It is unclear to me why the repetition rate and field were changed between different compounds. There should at least be some justification given for changing these parameters, as they could affect the measurement (for example, changing the repetition rate could artificially heat the sample, changing the strength of the field could affect the remagnetization dynamics). Additionally, it could be interesting to compare the measured speeds of de/remagnetization and amplitudes of the demagnetization in the dynamics in an attempt to compare the behavior of the materials in a quantitative way.

The proposed mechanism for the dynamics is spin reorientation accompanied by a precession. Authors believe that in order to have this precession/reorientation they must suppress the source of anisotropy of the Co or weaken its contribution. They support this with the fact that adding in too much Co causes the magnet not to flip.

The authors then search for a mechanism that would break the symmetry or favor a particular in-plane direction that they believe is necessary for switching induced by launching coherent magnons, they try to rule out certain sources of symmetry breaking by fabricating a wedge structure, but this is inconclusive. They also consider the symmetry to be broken by the gaussian laser pulse, but do not find any way to justify this either. Finally, they speculate that it could be due to spin currents that generate a spin transfer torque in the magnetization, however this seems somewhat unlikely since they are typically generated more efficiently over short timescales, which doesn't agree with the author's finding that the pulse duration does not affect the reversal of magnetization. They authors should probably at least discuss this.

Finally, the authors point out that they are demonstrating a new kind of all optical switching and rightly claim that its origins appear to be quite different from previous examples of all optical switching. This in my opinion gives the paper a strong case for publication in nature communications. The microscopic understanding is not completely proven, and should probably be discussed more to make this point clear, however the experimental results are solid.

Reviewer #3:

Remarks to the Author:

In their article entitled "In plane reorientation induced single laser pulse magnetization reversal in rare-earth based multilayer", Y. Peng et al study the possibility of achieving single-shot toggle-switching of magnetization in rare-earth based multilayers. The research field of all-optical switching has been very exciting and interesting over the last 15 years, showing that laser pulses can all-optically switch magnetization without the requirement of magnetic fields. It is important to point out that only in the last few years have research groups found the toggle-switching process

in materials not containing gadolinium (in particular, MRG Heusler alloys (Refs BAN20, DAV20b) and Tb/Co multilayers (Refs AVI19, AVI20)).

Y. Peng et al build particularly on the recent discovery of toggle-switching in Tb/Co multilayers, showing that multilayered stacks with the terbium and/or cobalt substituted with dysprosium, iron, platinum, boron and nickel also show the toggle-switching. The switching in these multilayers is different to the one usually found in GdFeCo and MRG, with the switching being independent of the pulse duration, and at higher laser fluences, the switching takes the form of rings. In GdFeCo, works have consistently found that the switching is very dependent on the pulse duration, and the switching always results in uniform spots (or at higher fluence, a spot of demagnetization surrounded by a single switched ring).

I congratulate the authors on studying a very extended set of materials (19 to be exact, shown in Table 1). They definitely show that this type of switching is a very general phenomenon. However, in my opinion, the authors do not present sufficient evidence to back up their physical interpretation of the underlying mechanism. Namely, that the switching evolves via an in-plane precessional reorientation of the magnetization. Also, I could not find a convincing physical explanation of why the multilayered structure seems to work so well for toggle-switching.

I understand that not all questions can be answered in a single article, but the two mentioned above are fundamental in explaining the actual switching. The reference to in-plane reorientation is heavily emphasized (even in the title), so it must be fully supported. In addition, the time-resolved measurements crucial for supporting the explanation were done with a DC magnetic field, so we do not actually know the dynamics of the switching process. I therefore cannot yet support publication of the manuscript in its present form in a high-impact journal such as Nature Communications. If the authors can provide more data and a coherent explanation for the observed results, I would be happy to reconsider my opinion.

Below, I present a series of questions/comments.

1. In Fig. 1-2, the static magnetization of the multilayer is imaged, showing that the switching gives rise to a single spot (at low fluence) and multiple rings (with the number scaling with fluence). I assume that the spot size is kept constant for these measurements, and the incident energy is merely increased. However, what happens if the energy is kept fixed, and the spot size is changed (focused or defocused)? I think this is important since the spot is Gaussian in profile – one would see a difference e.g. in the number of rings (or their width), if the Gaussian spot is very sharp or very broad.

2. In the paragraph after Fig. 2, the authors write “In the case of the Tb/Co and Tb/Fe multilayers, Tb is magnetic at room temperature only because of its proximity or contact with the Co or Fe layers.” This deserves more explanation. How is the magnetism induced in terbium e.g. by exchange? Is the magnetism in the terbium layer distributed equally, or is the face in contact with the cobalt more magnetized than in the center of the terbium nanolayer? How important is the ‘sharpness’ of the boundary between the nanolayers, and what effect would intermixing have?

3. In Fig 4 and Fig. S5, the measured signal undergoes a three-step process. 1) a very fast initial drop; 2) rapid recovery, and; 3) falling again. I can comprehend step 1 being a result of ultrafast demagnetization (presumably of the Co, since the probe mainly detects this), and step 2 resulting from recovery (usually seen in ferro- and ferri-magnets). However, the origin of step 3 is still a mystery. To try to help interpret and understand this step, can the authors please comment on the following:

a. What is the size of the probe spot used to obtain the results in Fig. 4? The authors write that this data was obtained by TR-MOKE images. Did the authors analyze pictures using a region of interest? The section on ‘Data Analysis’ only refers to calculations of the pump spot size.

b. What is the pump fluence for the signals shown in Fig. 4? The energy is given in Table 2, but the fluence should be clearly stated to provide direct comparison with the measurements in Fig 1,2,3 etc. If the fluences are not very comparable, it should be mentioned explicitly in the caption

so that the reader avoids quantitatively comparing the different signals.

c. Does the repetition rate play a role? The authors show 3 different repetition rates in Table 2 – was there a specific reason why different repetition rates were used?

d. The authors write that this cannot be attributed to multi-domain structure, since “the Kerr microscope images do not reveal any domain features”. Could the induced domains be smaller than the spatial resolution of the images?

e. Were additional images recorded like in Fig. S5, but with higher incident energy? I wonder if the authors can identify at what point in time the ringed structure (shown in Fig. 1b column 2 onwards) appears e.g. does the outermost ring appear first, or do all rings appear at the same time?

f. The authors claim that step 3 consists of in-plane reorientation of the magnetization during the switching process. However, the time-resolved results shown in Fig. 4 and Fig. S5 were obtained with a DC magnetic field (the strength and direction is not given), so the magnetization dynamics presented here do not correspond to actual switching. Could the magnetization be precessing about the applied DC field?

g. To ultimately resolve the direction of magnetization during the switching process, the authors could measure the in-plane component of magnetization. Did they try this? This would give the most convincing evidence of magnetization precession, tracking the in-plane and out-of-plane trajectory of the magnetization.

h. The magnetization precession claimed by the authors requires some form of coherence, and the authors suggest that this could arise from spin-polarized currents. However, I do not see any evidence of coherent rotation, since the time-resolved experiments show no clear oscillation in Fig. 4.

i. The result shown in Fig. S5 seems rather surprising, in the sense that step 3 persists with the signal staying close to zero for at least 800 ps. Can this be understood with the hypothesis of magnetization precession? Why would the magnetization “hang” at this point? And did the authors see any associated timescale of this plateau at longer time scales, since the magnetization must eventually precess either up or down (in the absence of magnetic field)?

4. To accurately model the system, I suppose the authors should start with transfer-matrix calculations to evaluate whether the transition metal layer absorbs more energy than the rare earth layer. If there is a substantial difference, this might help to explain the ringed features. If there is no major difference, the authors would then be more justified in using “averaged” parameters to characterize the 2-temperature model (connected to the next point).

5. The authors present a 2-temperature model to simulate their results. Critically, since they consistently study a multilayered structure, this raises a major question of whether a 2-temperature model even makes sense. It seems reasonable to assume an alloy has “averaged” parameters for the electron/lattice subsystems, but I do not see how this can be extended to a multilayered structure. At minimum, each layer will have a specific electron- and lattice subsystem, so this would give rise to two 2-temperature models. Do the authors consider this? Perhaps they do, but I could not see any discussion of it.

6. As I have already said, Table 1 is an impressive list of 19 materials that have been investigated. I find it a pity that the authors have shown time-resolved measurements in Fig. 4 for materials that should switch. It could be very useful to measure the time-resolved magnetization dynamics shown by a material not showing toggle-switching, as we could clearly compare the magnetization dynamics.

7. The authors do not present any clear explanation for why the switching seems to be so universally present in multilayered structures. Can the authors explain e.g. the different physics which seem to be shown by multilayers versus alloys? Why does Tb/Co show toggle-switching

while TbCo alloys do not? The group of Koopmans have nicely shown that the switching in Gd/Co multilayers seems to be better since there is clearly defined interfaces between the Gd and Co, supporting the ultrafast exchange scattering that leads to toggle-switching there. However, the switching here follows a different mechanism (as emphasized by the authors throughout the article).

8. The authors make no mention of the compensation point. Does the synthetic ferrimagnet require one, for toggle-switching? It is known that the experiment must be close to the compensation point to achieve toggle-switching in GdFeCo alloys, or below it for MRG – do similar requirements apply here? This might present a clue to explain the observed results, and indicate a constraint on the universal nature of the switching mechanism.

9. Can the authors give an error bar that should be associated with the layer thicknesses? The layer thicknesses are quoted to hundredths of nanometers (e.g. 1.06 nm, 1.78 nm etc in Table 2) - did the authors measure this with precision, or is it estimated from the sample growth procedure?

10. The authors refer consistently to permalloy – can the exact composition be provided? I could not see it written anywhere.

11. The title is rather confusing, as there are too many nouns without joining words. Overall, I recommend the authors to carefully proof-read the article, as there are a lot of typos. For example, in the abstract, I see:

a. "Since the first switching experiments carried on GdFeCo ferrimagnetic systems": should be "carried out"

b. "in this large family of (RE-TM) multilayer systems": the brackets should be removed

c. "Our results allow expanding the variety of materials...": this should be changed to "Our results expand the variety of materials..."

Dear Editor and Reviewers,

First of all, we would like to thank for the very positive comments of Reviewers 2 and 3 that point out that “

- *“our well written paper brings interesting new findings to the field”*
- *“the experimental results are solid”*
- *“the main findings are fundamentally new and would be of great interest to the community”*
- *“I congratulate the authors on studying a very extended set of materials (19 to be exact, shown in Table 1). They definitely show that this type of switching is a very general phenomenon”*
- *“This in my opinion gives the paper a strong case for publication in nature communications”.*

We believe that the novelty of our results cannot be questioned neither that this first extensive materials work show that this type of switching is a very general phenomenon. As a result, we think that our results will open the field for new researches and will motivate future theoretical and experimental studies.

Contrary to Reviewers 2 and 3, that acknowledge *“the demonstration of a new kind of all optical switching and rightly claim that its origins appear to be quite different from previous examples of all optical switching”*, Reviewer 1 is not convinced by the proof of this new mechanism. To further support our findings we have evidenced after the submission of the paper precession-like behaviour, that counteracts the statement *“this manuscript does not show any precessional motion”*. Additionally, we give arguments for the effective field precession considering precession at the scale of “magnetic” grains as supported by the MFM images.

The main criticism from Reviewer 1 arises from the lack of the unambiguous demonstration of precession due to the impossibility on our TR-MOKE set up to make measurements without applied field. *“The microscopic understanding is not completely proven”* but this is beyond the scope of the paper as also stated by the referee 3 *“I understand that not all questions can be answered in a single article”*. In the initial version of the paper, we discussed the processes that could be ruled out. In the revised version, we propose a complete explanation, that includes all specific ingredients supporting a precession-based reversal. We hope that this improved version, whose pertinence is based on the large number of samples studied, on the discussion of the ingredients to get the switching and on the possible proposed mechanisms will convince the reviewers.

In the following part of our rebuttal letter we tried to address carefully all the reviewers questions and comments. We hope the reviewers will agree with our detailed answers. Furthermore, we made complementary measurements to answer reviewers questions.

Reviewer #1 (Remarks to the Author):

Review report for “In plane reorientation induced single laser pulse magnetization reversal in rare-earth based multilayer”

The authors of Y. Peng et al. reported all-optical switching in Tb-based multilayers and claimed a new switching mechanism of in-plane reorientation. To argue the in-plane reorientation, they presented three major pieces of evidence: 1) switching is nearly independent on the pulse duration; 2) the switched domain shows concentric ring structures; 3) time-resolved dynamics show a gradual decreasing behavior. However, I am not convinced that these evidences undoubtedly support their claim.

1. pulse duration dependence

The authors showed that optical switching is nearly independent on the pulse duration. They claimed that the switching mechanism is not driven by the nonequilibrium electronic state and the state diagram of Fig. 1c is different from that of the optical switching in Gd-based system. I agree that the nonequilibrium state might be not a critical factor, but I am not convinced that this result can support the new switching mechanism.

We acknowledge that the independence on the pulse duration is not by itself a direct evidence of a distinct mechanism, but it constitutes an important difference with the conventional observations and understanding of AOS in GdFeCo. Moreover, the observed ring structure and dynamics as well as the material dependent results cannot be explained by a conventional AOS mechanism neither. It is for these reasons that we suggest that an alternative mechanism is needed to explain our results, even though many mechanisms are common with the conventional AOS (exchange of angular momentum, ultrafast demagnetization of sublattices...).

For example, J. Gorchon et al. reported an optical switching in GdFeCo alloy varying a pulse duration and showed that the critical fluence increases slightly with a wide pulse duration of 55 fs ~ 15 ps [Fig. 4 of PRB 94, 1844006 (2016)]. Therefore, in my opinion, a similar switching diagram (Fig. 1c) has been observed in a conventional Gd-based system, in which the switching mechanism is known to be related to a difference in the demagnetization time of the two sub-lattices.

Thanks to the referee to bring this comparison. However, we do not agree that the two switching diagrams are similar. To support our statement we do a comparison with the Figure 4 of the same reference J. Gorchon et al, PRB 94, 1844006 (2016). While it is difficult to conclude due to large error bars in the case of the Gd₂₄FeCo concentration, in the case of Gd₂₇FeCo concentration, the critical fluence increases by 50% when changing the pulse duration from ~100 fs to ~10 ps. In contrast in our data the critical fluences vary by less than 5%. Moreover, by increasing the pulse duration, F_C eventually crosses the multidomain threshold $F_{\text{multidomain}}$ which is mostly pulse duration independent, and AOS cannot be obtained anymore. We could not observe such a crossing of the multidomain and critical thresholds in our samples. These pulse duration dependencies for GdFeCo samples have also been reported in [WEI21] with even lower error bars and this results further support our claims. The fact that, in our new paper F_C does not depend on the pulse duration, is a completely new finding.

FIG. 4. Critical fluence F_C for AOS and multidomain states as a function of (a) the initial temperature of the sample for $\Delta t = 55$ fs laser pulses and (b) the laser pulse duration at room temperature. Solid lines are guides for the eyes. The blue dashed lines are a calculation of the fluence needed to make the lattice reach T_C (see text). MOKE images in (a) show the typical result in each fluence range. From bottom to top: No switch (ultrafast demagnetization), AOS, and multidomain state. The vertical dashed lines in (b) show the limits for observation of AOS in each sample. The right hand image shows the fully demagnetized state obtained for a $\Delta = 16$ ps pulse of ~ 1.85 mJ/cm^2 on sample $\text{Gd}_{27}\text{FeCo}$.

2. Concentric ring structure

The concentric ring structure of the switched domains is indeed different from that of Gd-based system, but it looks similar to that of Iron garnet system. C. S. Davies et al. reported the concentric ring structure after the optical switching in iron garnet [PRL 122, 027202 (2019)]. This ring structure was explained by nonlinear magnetization precession, which is controlled by laser heating and external magnetic field. According to Davies et al., to have the magnetization precession, an external in-plane field should exist. However, the optical switching of this manuscript was done without an external field. The authors of this manuscript tried to find the physical origin that acts as an effective in-plane field but could not identify it. Therefore, except that the ring structure is similar to that of iron garnet, the authors could not provide a plausible explanation for how the ring structure is driven by in-plane reorientation.

The reviewer is right, we do not apply a field in plane which makes our finding new. We tried to find a physical origin of an effective in-plane field to bridge our results with those obtained in iron garnet samples. Actually, the response is not obvious. However, precession could happen starting from a tilted anisotropy axis and having an anisotropy axis in the plane. This anisotropy axis could be local, at the scale of the grain while total anisotropy, averaged over the layer plane is zero. The precession would occur around this local anisotropy and when the equatorial plane of the film is crossed, reversal occurs. We know that fluctuations of the perpendicular anisotropy exist in our material. We have performed additional MFM measurements and we clearly see contrast in the region where magnetization is saturated. Those fluctuations are clearly not correlated to the roughness of the layers.

We also know that TM-RE alloys are amorphous and the FM layers are polycrystalline or textured. Therefore we expect varying local directions of anisotropy could promote local precession of magnetization. We support a local precession on the basis of the analysis of magnetic force microscopy images. All these arguments have been added to the text.

3. Time-resolved dynamics

The authors showed an unexpected decrease in magnetization after a few ps and claimed that it is due to the in-plane reorientation. Although this decrease cannot be explained by simple temperature analysis, I am not convinced that it comes from the in-plane reorientation. When the laser heating induces a sudden change in magnetic anisotropy and causes an in-plane reorientation, the magnetization dynamics show multiple precessions [PRB 97, 014422 (2018)]. However, Fig. 4 of this manuscript does not show any precessional motion. Why? In addition, time-resolved dynamics were done with an external field, whereas optical switching was done without a field. With such a different condition, how one can be sure that the same mechanism governs both experiments?

To reset at every step the magnetic state of our system, magnetization dynamics have been recorded under perpendicular external applied field. In our setup, it is not possible to measure the dynamics without an applied field. As a result, if a precession occurs, that will be around the direction of the effective magnetic field (including anisotropy field + external field). In our case, we see a demagnetization + remagnetization behaviour first, and then the magnetization falls towards the in plane configuration in a 100-200 ps timescale, which could be related to the precession frequency. The lack of oscillations can be explained by the large damping of the materials.

In some cases, we have been able to identify oscillations. In the following, an example is given. We did not do a systematic study of those oscillations, but we know that they exist. They are in agreement with a strongly damped reorientation of the anisotropy. A paragraph has been added to the text to show that, even if the conditions are not optimal, precession can be measured. A figure has been added in supplementary materials.

Figure : Dynamics of a V/Ta(5)Pt(5)[Co(2nm, wedge)/(Co60Tb40)(4)]x3Ta(5)Pt(1) multilayer. The dynamics are recorded using different external applied magnetic fields.

In summary, the novelty and clarity of this work are not high enough to be published in a high-impact journal of Nature Communications. Therefore, I recommend authors collect more clear evidence to support their claim.

We believe that the novelty of our results cannot be questioned and we hope that the additional arguments brought by our answers here above are convincing. As a reminder, we show AOS in a new class of Gd-free multilayers, showcasing toggling concentric rings without the use of an in-plane magnetic field.

However, we acknowledge that the lack of clarity indicated by the Reviewer 1 appreciated comes from the challenge in pinpointing the exact mechanism responsible for the observations. In the first version of the paper, we mainly discussed the processes that could be ruled out. In the revised version of the paper, we added a more speculative explanation that is coherent with all the ingredients to have the reversal by precession. Hopefully, Reviewer 1 will appreciate the numerous samples we studied, the unique ingredients that we have highlighted and discussed, as well as the new possible mechanism we have proposed.

Reviewer #2 (Remarks to the Author):

In general, it is a well written paper, placed well in relation to recent works, that brings interesting new findings to the field. The explanation of the results is somewhat weak, in that the authors propose a few mechanisms but fail to conclusively support one. In addition, some of the supporting data is scattered, and seems to vary too many parameters to have any kind of systematic significance. However, I do believe that the main findings are fundamentally new and would be of great interest to the community. Therefore, I would like the authors to address the following concerns and possibly re-organize the manuscript to make it more readable before recommending the article for publication in Nature Communications.

We thank the reviewer for appreciating the manuscript and its interest for the field. We also acknowledge the “weakness” of our previous explanation, which we have tried to improve in the new version.

The paper shows very systematically that all optical switching with a single laser pulse with up to 10 ps of duration is a relatively ubiquitous phenomena and that it seems to depend on certain material properties of the materials involved. It would be helpful to state this more clearly in the introduction/conclusion.

This point has been included in the conclusion of the paper.

One very interesting finding of the paper is that the flipping of the magnetic order is not dependent on pulse duration (up to 10ps) or peak energy, but rather the total amount energy deposited in the system. The authors follow up on this with a 2T model, suggesting that it is the lattice temperature, rather than the electron temperature that is the determining factor in the magnetic reversal process. It is unclear why the authors did not attempt to use a 3 temperature model, since excitation of the spins could also be a relevant timescale. The authors should at least justify the fact that they did not use a 3T model here.

Three temperature models add even more variables, which are not generally known. Moreover, it is not completely obvious how to define spin temperatures in ferrimagnets. What is known is that spins typically follow closely the temperature of electrons, whereas the lattice is generally much slower. Because we do not see major changes with pulse duration, we wanted to show numerically what is already intuitively known: that phonon peak temperatures depend on total energy absorption, whereas peak electron temperatures depend not only on the total energy absorption, but also on the rate of energy absorption. These numerical simulations support our claim that peak electron temperature cannot govern the magnetization reversal, but that the peak phonon temperature might. Adding a spin bath to the calculation will not change this statement. We have added a short sentence in the text to clarify this point.

The authors state that the importance of the transfer of energy to the phonon is justified by the strong spin orbit coupling present for Tb or Dy, contrary to Gd. However, they also state that a much more robust toggle switching is obtained when Tb is replaced by ferrimagnetic alloys with thicknesses up to few tens of nanometers and for laser pulses as long as 12 ps. They do not offer an explanation for why this is, and the spin-orbit coupling explanation would clearly not work here, since typically these alloys would have a weaker spin-orbit coupling.

Thanks to the referee for this remark. To be more explicit let's compare directly Co/Tb and Co/CoTb multilayers. In both multilayers, the reversal occurs when the composition of the multilayer is close to the total magnetization compensation. This means that the total moment of Tb is close to the total

moment of Co. Therefore, the contents of Tb are very close in the Co/Tb and Co/CoTb samples, we expect the total spin-orbit coupling not to differ too much. In contrast, the magnetic properties are much well defined in the CoTb alloy with respect to the Tb single layer and the Co/Tb interfaces.

We have modified this part of the paper to explain what we mean by “more robust” : we demonstrate that the toggle switching obtained in Co/Tb is robust for laser pulses as long as 12 ps and with respect to layer thickness (up to few tens of nanometers) or TM / RE chemical nature when Tb is replaced by ferrimagnetic alloys.

In Figure 4 and Table 2 the authors plot the time dynamics for several different compounds. However, the experimental conditions were not the same for the different compounds, which is why they have to be given in Table 2. It is unclear to me why the repetition rate and field were changed between different compounds. There should at least be some justification given for changing these parameters, as they could affect the measurement (for example, changing the repetition rate could artificially heat the sample, changing the strength of the field could affect the remagnetization dynamics). Additionally, it could be interesting to compare the measured speeds of de/remagnetization and amplitudes of the demagnetization in the dynamics in an attempt to compare the behavior of the materials in a quantitative way.

In order to have the highest signal to noise, a high laser repetition rate is used. However, for some samples, repeated heating is causing the sample to be burned due to heat accumulation. Therefore, a compromise has to be found to have a reasonable acquisition time, without too much heat accumulation. Depending on the sample's magneto-optical contrast, we had to change this parameter for different experiments. In terms of the magnetic field, as the referee said, the magnitude of the magnetic field affects how fast the magnetic moments remagnetize on the long delays (above 10 ps), but should not affect the short time dynamics.

The proposed mechanism for the dynamics is spin reorientation accompanied by a precession. Authors believe that in order to have this precession/reorientation they must suppress the source of anisotropy of the Co or weaken its contribution. They support this with the fact that adding in too much Co causes the magnet not to flip. The authors then search for a mechanism that would break the symmetry or favor a particular in-plane direction that they believe is necessary for switching induced by launching coherent magnons, they try to rule out certain sources of symmetry breaking by fabricating a wedge structure, but this is inconclusive. They also consider the symmetry to be broken by the gaussian laser pulse, but do not find any way to justify this either. Finally, they speculate that it could be due to spin currents that generate a spin transfer torque in the magnetization, however this seems somewhat unlikely since they are typically generated more efficiently over short timescales, which doesn't agree with the author's finding that the pulse duration does not affect the reversal of magnetization. They authors should probably at least discuss this.

In the first version of the paper, we discussed the processes that could be ruled out. In the revised version, we add a more speculative explanation that is coherent with all the ingredients to have the reversal by precession.

Finally, the authors point out that they are demonstrating a new kind of all optical switching and rightly claim that its origins appear to be quite different from previous examples of all optical switching. This in my opinion gives the paper a strong case for publication in nature communications. The microscopic understanding is not completely proven, and should probably be discussed more to make this point clear, however the experimental results are solid.

We thank the reviewer for acknowledging the interest of our paper and agreeing in that we are demonstrating a new mechanism. As stressed by the referee, a strong point of the paper is the

demonstration that this new reversal could be observed using many different RE alloys multilayers. As suggested by the reviewers, in the revised version, we have tried to improve the discussion around the microscopic understanding and clarified various of the points raised by the referee.

Reviewer #3 (Remarks to the Author):

In their article entitled!! “In plane reorientation induced single laser pulse magnetization reversal in rare-earth based multilayer”, Y. Peng et al study the possibility of achieving single-shot toggle-switching of magnetization in rare-earth based multilayers. The research field of all-optical switching has been very exciting and interesting over the last 15 years, showing that laser pulses can all-optically switch magnetization without the requirement of magnetic fields. It is important to point out that only in the last few years have research groups found the toggle-switching process in materials not containing gadolinium (in particular, MRG Heusler alloys (Refs BAN20, DAV20b) and Tb/Co multilayers (Refs AVI19, AVI20)).

Y. Peng et al build particularly on the recent discovery of toggle-switching in Tb/Co multilayers, showing that multilayered stacks with the terbium and/or cobalt substituted with dysprosium, iron, platinum, boron and nickel also show the toggle-switching. The switching in these multilayers is different to the one usually found in GdFeCo and MRG, with the switching being independent of the pulse duration, and at higher laser fluences, the switching takes the form of rings. In GdFeCo, works have consistently found that the switching is very dependent on the pulse duration, and the switching always results in uniform spots (or at higher fluence, a spot of demagnetization surrounded by a single switched ring).

I congratulate the authors on studying a very extended set of materials (19 to be exact, shown in Table 1). They definitely show that this type of switching is a very general phenomenon. However, in my opinion, the authors do not present sufficient evidence to back up their physical interpretation of the underlying mechanism. Namely, that the switching evolves via an in-plane precessional reorientation of the magnetization. Also, I could not find a convincing physical explanation of why the multilayered structure seems to work so well for toggle-switching.

I understand that not all questions can be answered in a single article, but the two mentioned above are fundamental in explaining the actual switching. The reference to in-plane reorientation is heavily emphasized (even in the title), so it must be fully supported. In addition, the time-resolved measurements crucial for supporting the explanation were done with a DC magnetic field, so we do not actually know the dynamics of the switching process. I therefore cannot yet support publication of the manuscript in its present form in a high-impact journal such as Nature Communications. If the authors can provide more data and a coherent explanation for the observed results, I would be happy to reconsider my opinion.

Below, I present a series of questions/comments.

1. In Fig. 1-2, the static magnetization of the multilayer is imaged, showing that the switching gives rise to a single spot (at low fluence) and multiple rings (with the number scaling with fluence). I assume that the spot size is kept constant for these measurements, and the incident energy is merely increased. However, what happens if the energy is kept fixed, and the spot size is changed (focused or defocused)? I think this is important since the spot is Gaussian in profile – one would see a difference e.g. in the number of rings (or their width), if the Gaussian spot is very sharp or very broad.

In order to respond to reviewer’s comment, we made the experiment to change the spot size.

As can be seen in this figure, the size of the spot does not affect the formation and disappearance of the rings, only the size of the magnetic domain (the radius of the ring) as well as the local energy, but the fluences at the domain walls do not change.

2. In the paragraph after Fig. 2, the authors write “In the case of the Tb/Co and Tb/Fe multilayers, Tb is magnetic at room temperature only because of its proximity or contact with the Co or Fe layers.” This deserves more explanation. How is the magnetism induced in terbium e.g. by exchange? Is the magnetism in the terbium layer distributed equally, or is the face in contact with the cobalt more magnetized than in the center of the terbium nanolayer? How important is the ‘sharpness’ of the boundary between the nanolayers, and what effect would intermixing have?

The magnetic properties of RE/TM multilayers have been extensively studied in the past. For example in Journal of Applied Physics 64, 5748 (1988), Fe/Tb multilayers made of thick Fe and Tb have been studied. It was namely shown that the magnetic system is made of ferrimagnetically coupled Tb-Fe. The magnetism in Tb is induced by exchange, the magnetism in Tb is not distributed equally, the face in contact with the cobalt is more magnetized and antiferromagnetically coupled. The ‘sharpness’ of the boundary between the layers can play a role and is controlled by the deposition conditions. The reference has been added to the text. We did not detailed all these considerations in order to keep a light paragraph and to simplify the overall message of the paper. We think that the reader will find this sentence reasonable considering that the Tb is not magnetic at the room temperature and will always be able to refer to the reading of the article if needed.

3. In Fig 4 and Fig. S5, the measured signal undergoes a three-step process. 1) a very fast initial drop; 2) rapid recovery, and; 3) falling again. I can comprehend step 1 being a result of ultrafast demagnetization (presumably of the Co, since the probe mainly detects this), and step 2 resulting from recovery (usually seen in ferro- and ferri-magnets). However, the origin of step 3 is still a mystery. To try to help interpret and understand this step, can the authors please comment on the following:

We do not believe that it is a mystery. We clearly state and show that step 3 is the reorientation of the magnetization in plane by the loss of anisotropy in the CoTb and so also of Co (since Co is perpendicular through its exchange with the CoTb layers). We will give the answers to all the question of the reviewer in the following concerning this point.

a. What is the size of the probe spot used to obtain the results in Fig. 4? The authors write that this data was obtained by TR-MOKE images. Did the authors analyze pictures using a region of interest? The section on ‘Data Analysis’ only refers to calculations of the pump spot size.

Yes, the TR-MOKE shown in figure 4 is extracted from TR-MOKE images and the region of interest is the central region, where the fluence is the highest. It corresponds to a diameter of 2 μ m. The size of the region of interest has been added to the figure caption.

b. What is the pump fluence for the signals shown in Fig. 4? The energy is given in Table 2, but the fluence should be clearly stated to provide direct comparison with the measurements in Fig 1,2,3 etc. If the fluences are not very comparable, it should be mentioned explicitly in the caption so that the reader avoids quantitatively comparing the different signals.

Stacks	Composition	Field (Oe)	Fluence(mJ/cm ²)	Repetition rate(kHz)
[Tb/Co] ₅ multilayer	t(Tb) = 1.06 nm t(Co) = 1.78 nm	550	6.32	0.2
[TbCo/Co] ₃ multilayer	t(Tb ₄₀ Co ₆₀) = 4 nm t(Co) = 2.10 nm	610	6.52	10
[DyCo/Co] ₃ multilayer	t(Dy ₃₀ Co ₇₀) = 4 nm t(Co) = 2.18 nm	570	3.83	1
TbCo/Co/TbCo trilayer	t(Tb ₃₂ Co ₆₈) = 4 nm t(Co) = 2.24 nm	690	3.26	1

We thank the reviewer for his/her comment. We changed the energy column to fluence. The table has been changed in the paper. It has been moved to supplementary material because a new figure has been added to the text of the manuscript.

c. Does the repetition rate play a role? The authors show 3 different repetition rates in Table 2 – was there a specific reason why different repetition rates were used?

A high laser repetition rate allows better contrast to be detected, but it also heats up the sample too much, causing the sample to be burned due to heating accumulation. Therefore, while ensuring that the sample is not burned, in order to obtain better contrast, the repetition rate of the laser will be appropriately increased. A sentence has been added to the caption of the table.

d. The authors write that this cannot be attributed to multi-domain structure, since “the Kerr microscope images do not reveal any domain features”. Could the induced domains be smaller than the spatial resolution of the images?

If the domains have a lateral extension less than the spatial resolution, we could indeed not see them. However, in our experiment, we apply a field perpendicular to film plane in order to reset the magnetization in its initial state. At long timescales, above typically 50ps, this applied field will prevent the formation of domains. Therefore, the decrease of magnetization that is observed can only be linked to a reorientation of magnetization in plane.

e. Were additional images recorded like in Fig. S5, but with higher incident energy? I wonder if the authors can identify at what point in time the ringed structure (shown in Fig. 1b column 2 onwards) appears e.g. does the outermost ring appear first, or do all rings appear at the same time?

The dynamics in Tb/Co multilayer (in figure 4 and figure S5) has been done with fluence of 6.32 mJ/cm², with which the ring structure in figure S2-a column 2 could be observed. However, the switching as well as the formation of ring structure have not been observed up to a time delay of 2ns. We believe this is linked to the applied field needed to reset the multilayer magnetic state. As a result, we could not identify which ring appears the first.

f. The authors claim that step 3 consists of in-plane reorientation of the magnetization during the switching process. However, the time-resolved results shown in Fig. 4 and Fig. S5 were obtained with a DC magnetic field (the strength and direction is not given), so the magnetization dynamics presented here do not correspond to actual switching. Could the magnetization be precessing about the applied DC field?

The reviewer is right. To reset the magnetic state of the multilayer, dynamics have been recorded under perpendicular applied field. In our setup, it is not possible to measure the dynamics without an applied field. As a result, if a precession occurs, it will be around the direction of the effective magnetic field (anisotropy field + external field). In our case, we see a demagnetization + remagnetization behaviour first, and then the magnetization falls towards the plane in a 100-200 ps timescale, which could be related to the precession frequency. The lack of oscillations can be explained by the large damping of the materials.

In some cases we could find some oscillations. In the following, an example is given. We did not do a systematic study of those oscillations, but we know that they exist. They are in agreement with a strongly damped reorientation of the anisotropy. We added these curves in the supplementary material.

Figure : Dynamics of a V/Ta(5)Pt(5)[Co(2nm, wedge)/(Co60Tb40)(4)]x3Ta(5)Pt(1) multilayer. The dynamics are recorded using different applied fields.

g. To ultimately resolve the direction of magnetization during the switching process, the authors could measure the in-plane component of magnetization. Did they try this? This would give the most convincing evidence of magnetization precession, tracking the in-plane and out-of-plane trajectory of the magnetization.

The reviewer is right. It would be of interest to measure the in plane component of the magnetization. However, such a measurement would require the application of a magnetic field in plane in order to insure an in-plane saturation of the magnetization which would be needed to detect a signal. This field would once again modify the dynamics making a direct comparison with experiments without any field applied difficult. Nevertheless, some precession have been observed when varying the applied magnetic field perpendicular therefore confirming their existence (see above).

h. The magnetization precession claimed by the authors requires some form of coherence, and the

authors suggest that this could arise from spin-polarized currents. However, I do not see any evidence of coherent rotation, since the time-resolved experiments show no clear oscillation in Fig. 4.

We agree with the reviewer on this point . On the basis of local magnetic force microscopy images, we find arguments to support that fact that the precession is local and that coherence over the size of the laser spot is not required. This part of the paper has been re-written and the spin currents, although we know they exists, have not been included in the text to keep the message readable

i. The result shown in Fig. S5 seems rather surprising, in the sense that step 3 persists with the signal staying close to zero for at least 800 ps. Can this be understood with the hypothesis of magnetization precession? Why would the magnetization “hang” at this point? And did the authors see any associated timescale of this plateau at longer time scales, since the magnetization must eventually precess either up or down (in the absence of magnetic field)?

The plateau could be seen up to 1.65ns. We are limited by the delay line of the set up to go longer. The long lasting state could be related to the long dissipation timescale. The samples are grown on glass which is notoriously a bad heat conductor. Therefore we believe that thermal diffusion may require longer than a few ns to cool down the sample and retrieve back the out-of-plane magnetic anisotropy.

4. To accurately model the system, I suppose the authors should start with transfer-matrix calculations to evaluate whether the transition metal layer absorbs more energy than the rare earth layer. If there is a substantial difference, this might help to explain the ringed features. If there is no major difference, the authors would then be more justified in using “averaged” parameters to characterize the 2-temperature model (connected to the next point).

In this example, we calculated the absorption using a transfer-matrix approach as proposed by the referee. The refractive index of $\text{Co}_{67}\text{Tb}_{32}$ was extracted from Ciuciulkate PRMat 4, 104418 (2020) for $\text{Co}_{71}\text{Tb}_{29}$. As we can see from the calculation, the absorption in Co and CoTb alloys are very similar therefore justifying the use of averaged parameters within our 2-temperature model. Furthermore, the difference in electronic temperatures induced during the light absorption would be leveled within the different layers within few tens of femtosecond since the excited electrons moves at a speed around 1 nm/fs.

5. The authors present a 2-temperature model to simulate their results. Critically, since they consistently study a multilayered structure, this raises a major question of whether a 2-temperature model even makes sense. It seems reasonable to assume an alloy has “averaged” parameters for the electron/lattice subsystems, but I do not see how this can be extended to a multilayered structure. At minimum, each layer will have a specific electron- and lattice subsystem, so this would give rise to two 2-temperature models. Do the authors consider this? Perhaps they do, but I could not see any discussion of it.

We agree with the referee. This was not considered in the model. However, we think that this is probably not too important since, we focused on the slow dynamics.

6. As I have already said, Table 1 is an impressive list of 19 materials that have been investigated. I find it a pity that the authors have shown time-resolved measurements in Fig. 4 for materials that should switch. It could be very useful to measure the time-resolved magnetization dynamics shown by a material not showing toggle-switching, as we could clearly compare the magnetization dynamics.

We did not concentrate our effort to measure samples that do not show the toggle switching. We wanted to compare the dynamics of the samples that show toggle switching to the one of Gd-based samples to show a completely different dynamics .

7. The authors do not present any clear explanation for why the switching seems to be so universally present in multilayered structures. Can the authors explain e.g. the different physics which seem to be shown by multilayers versus alloys? Why does Tb/Co show toggle-switching while TbCo alloys do not? The group of Koopmans have nicely shown that the switching in Gd/Co multilayers seems to be better since there is clearly defined interfaces between the Gd and Co, supporting the ultrafast exchange scattering that leads to toggle-switching there. However, the switching here follows a different mechanism (as emphasized by the authors throughout the article).

In the revised version, we add a more speculative explanation that is coherent with all the ingredients to have the reversal by precession and why a TM layer is needed to have the reversal.

8. The authors make no mention of the compensation point. Does the synthetic ferrimagnet require one, for toggle-switching? It is known that the experiment must be close to the compensation point to achieve toggle-switching in GdFeCo alloys, or below it for MRG – do similar requirements apply here? This might present a clue to explain the observed results, and indicate a constraint on the universal nature of the switching mechanism.

Except samples 8 and 14 in table 1, the single switching has only been observed in Co-rich region. It is worth noting that the thicknesses of TbCo alloy layer in a $[\text{TbCo}(7.5 \text{ nm})/\text{Co}]_3$ multilayer and Pt/Co/TbCo(16 nm) are very thick, and only partial switching could be obtained. Nothing seems to be related to the compensation temperature as asked by the referee

9. Can the authors give an error bar that should be associated with the layer thicknesses? The layer thicknesses are quoted to hundredths of nanometers (e.g. 1.06 nm, 1.78 nm etc in Table 2) - did the authors measure this with precision, or is it estimated from the sample growth procedure?

The variation of the layer thickness along the wedge has been calibrated using a thick Cu layer. The wedge-shaped Cu sample spanned of 50 mm. At position 25mm, the thickness was targeted to be 200 nm. The thickness of Cu has been measured using atomic force microscopy. The variation of the thickness of Cu was fitted as $t_{Cu} = 110.9 + 3.83x$, where x is the position in mm, as shown in figure below. Thus, the thickness t in the wedge can be obtained by $t(x) = \frac{110.9}{200} \times t_{25mm} + t_{25mm} \times \frac{3.83}{200}x$, where t_{25mm} is the expected thickness at 25 mm. The uncertainty of the thickness is thus $\Delta t(x) = t_{25mm} \times \frac{3.83}{200} \Delta x$ where Δx is the uncertainty in the position in the sample. Δx was estimated to be 0,1mm. Therefore, $\Delta t(x) \approx 0.002 t_{25mm}$. Since wedges have typical $t_{25mm} < 5nm$, $\Delta t(x) < 0.01$. That is the reason why the layer thicknesses are quoted to hundredths of nanometers.

This discussion has been added in the supplementary material

10. The authors refer consistently to permalloy – can the exact composition be provided? I could not see it written anywhere.

Permalloy is a Fe80Ni20 atomic composition alloy. This composition has been added to the text.
The CoFeB atomic composition is Co40Fe40B20. This composition has been added to the text.

11. The title is rather confusing, as there are too many nouns without joining words. Overall, I recommend the authors to carefully proof-read the article, as there are a lot of typos. For example, in the abstract, I see:

- “Since the first switching experiments carried on GdFeCo ferrimagnetic systems”: should be “carried out”
- “in this large family of (RE-TM) multilayer systems”: the brackets should be removed
- “Our results allow expanding the variety of materials...”: this should be changed to “Our results expand the variety of materials...”

We have carefully proof-read the article and corrected several typos.

Reviewers' Comments:

Reviewer #1:

Remarks to the Author:

Review report for the 1st revision of "In plane reorientation induced single laser pulse magnetization reversal in rare-earth based multilayer"

1. Pulse duration dependence: I raised a point that the pulse-width dependence is not significant even in a conventional GdFeCo system. The authors responded that a critical fluence for AOS in the Gd₂₇FeCo system increases by a factor of two when the pulse width increases from 0.1 ps to 13 ps [PRB 94, 1844006 (2016)]. Now I agree that the pulse-width independent AOS in this work is somewhat different from the pulse-width dependent AOS in GdFeCo system. I can also agree that the switching mechanism is related to the lattice temperature rising rather than non-equilibrium electron temperature.

2. Concentric ring structure: I raised a point that there is no direct evidence to relate the concentric ring structure to the in-plane rotation. The authors still do not provide supporting evidence but continue their claim based on speculations.

3. Time-resolved dynamics: I raised a point that if the in-plane rotation is responsible for the dynamics there should be a precession motion. The authors responded that a lack of the precessional motion would be due to a large damping. But I can easily find several papers that shows a precession motion of GdFeCo (for example, PRB 73, 220402(R) (2006)). Therefore, the authors's response is not satisfactory. The authors showed an addition TRMOKE data and claimed that some of data shows a precessional motion. But the precession motion should have a well-defined frequency and relaxation motion. I do not agree that this data represents a precessional motion. I also raise a point that TRMOKE was done with DC magnetic field whereas AOS was done without magnetic field. The authors responded that the DC magnetic field is necessary to reset the magnetization during TRMOKE experiment and it is not so critical to discuss the AOS mechanism. I can agree that the DC field may not be so important to discuss the mechanism.

Overall, I can agree with some points, but still not convinced that the in-plane rotation is responsible for the AOS. I understand that a single paper cannot answer all questions, but the key claims should be convincing enough. The observation itself is interesting but is not significant enough to be accepted in a high impact journal without a convincing explanation for the mechanism. I would like to give one more chance for the authors to provide more convincing evidence for the in-plane rotation.

1. Please provide the estimation of lattice temperature as a function of the pump fluence. Please provide the measurement of anisotropy energy as a function of temperature. Then, show that the lattice temperature at the critical fluence matches to the temperature where the anisotropy changes from perpendicular anisotropy to in-plane anisotropy.

2. If a decrease of magnetization signal at ~100 ps in TRMOKE data is caused by the in-plane rotation, this behavior will depend on the pump fluence. When the pump fluence is smaller than the critical fluence, the direction of magnetization remains to the out-of-plane direction so that there should be no in-plane rotation. In another word, the decrease of magnetization at ~100 ps will have a threshold behavior with the pump fluence. Please demonstrate this point.

Reviewer #2:

Remarks to the Author:

The authors have responded well to all of my concerns. I can now recommend this article for publication.

Reviewer #3:

Remarks to the Author:

The authors have mostly responded well to the questions of the reviewers, and made appropriate improvements to the manuscript.

However, I am still not convinced that the authors satisfactorily explain why magnetization precession occurs. The authors say when answering Reviewer 1 that "this response is not obvious", and now argue that fluctuations in the anisotropy's orientation is responsible. Further details are missing in their explanation.

In lines 319-321, the authors write that the direction of anisotropy fluctuates because of the polycrystalline layer. Why is this? Can the authors at least provide a reference, and ideally explain why this occurs?

How large is this random fluctuation of anisotropy i.e. does it deviate from out-of-plane by a small fraction, or does it vary from in-plane to out-of-plane (over 90degrees)?

How does the laser heating cause the anisotropy to temporarily go in-plane, in order to provide the necessary torque for precession?

The authors write "the persistence of the magnetization of the TM FM layer due to its high T_c ". Are the authors claiming that the magnetization at the interface (on the RE side) is destroyed, and the magnetization within the FM layer persists? If so, what is the T_c of the FM layer, and how much of its magnetization actually quenches? An estimate should be given.

Linked to this, the authors present interesting new data in Fig. S10, which could actually make a major step towards proving the precessional behavior. In Fig. S10, the magnetic field is changed in strength, and we see a distinct threshold-like behavior. With 377mT and 557mT, the curves are roughly the same (just changed in amplitude), whereas with 196mT, the downward slope becomes much longer (taking up to 200ps to level off). Further description and systematic parameters must be provided here. Were these measurements done with all parameters the same e.g. fluence? If the magnetic field is flipped, do we see symmetric behaviour? Can the Kerr signal be actually calibrated, so we can see M_z in terms of 100%, 0% etc (from the imaging, I think this should be straightforward)? And can the authors provide several more curves with different field, as this would show a clear threshold behavior (consistent with precessional dynamics)?

Overall, I think the article has benefitted from the review process so far. However, questions still remain. If the authors can provide a suitable response, showing better the precessional dynamics and explaining its origin, the article should be suitable for publication in Nature Communications.

REVIEWER COMMENTS

Reviewer #1 (Remarks to the Author):

Review report for the 1st revision of "In plane reorientation induced single laser pulse magnetization reversal in rare-earth based multilayer"

1. I can also agree that the switching mechanism is related to the lattice temperature rising rather than non-equilibrium electron temperature.

We are happy to have convinced referee 1 that the switching mechanism is related to the lattice temperature rising rather than non-equilibrium electron temperature.

2. Concentric ring structure: I raised a point that there is no direct evidence to relate the concentric ring structure to the in-plane rotation. The authors still do not provide supporting evidence but continue their claim based on speculations.

As written by referee 1, the end of his/her comment on this point was this one... "The authors of this manuscript tried to find the physical origin that acts as an effective in-plane field but could not identify it. Therefore, except that the ring structure is similar to that of iron garnet, the authors could not provide a plausible explanation for how the ring structure is driven by in-plane reorientation".

In fact, we have a plausible explanation of the ring structure driven by the in-plane reorientation as it is written in the text since we have found the origin of the effective field. By the use of the MFM analysis of the micromagnetic configuration, we really went deeper in the analysis of the origin of the effective field around which the local magnetization will precess. This takes place at a very local scale, the scale of the size of a grain or couple of grains. On this point, we think that the referee missed our detailed explanation.

3. Time-resolved dynamics: I raised a point that if the in-plane rotation is responsible for the dynamics there should be a precession motion. The authors responded that a lack of the precessional motion would be due to a large damping. But I can easily find several papers that shows a precession motion of GdFeCo (for example, PRB 73, 220402(R) (2006)). Therefore, the authors's response is not satisfactory.

As shown in Appl. Phys. Lett. 105, 072411 (2014), doping CoFe with Gd can decrease the spin-orbit interaction which in turn decreases the damping of the material. So we fully understand that precession motion can be observed in CoFeGd since damping can be lower than in pure CoFe. However, we do not understand why referee 1 gives the example of CoFeGd to say that we should not observe precession in Tb or Dy based materials. Indeed, it is shown in Phys. Rev. Lett. 102, 257602, that the damping of NiFe strongly depends on the doping of rare earth : changing Gd to Tb or Dy, the damping increases by a factor of 10. So all those papers show that damping is very high Tb and Dy based materials. Furthermore, a local distribution of anisotropy could also lead to an enhancement of the damping parameters which makes difficult the observation of magnetization precession Malinowski et al. Appl. Phys. Lett. 94, 102501 (2009)

Based on this analysis, we do not understand the arguments of referee 1 and we hope that our reply is satisfactory.

The authors showed an addition TRMOKE data and claimed that some of data shows a precessional motion. But the precession motion should have a well-defined frequency and relaxation motion. I do not agree that this data represents a precessional motion.

We do not agree with referee 1 on this point . Indeed, a well defined frequency can not be observed in strongly damped materials. This has been for example shown in J. Phys. D: Appl. Phys. 55 (2022) 175001. In this study, it is shown that for in plane magnetized TbCo2/FeCo multilayers, oscillations were strongly damped and became undistinguished after two (samples A,C) or even one period (samples B,D). This behaviour support undoubtedly our statement.

I also raise a point that TRMOKE was done with DC magnetic field whereas AOS was done without magnetic field. The authors responded that the DC magnetic field is necessary to reset the magnetization during TRMOKE experiment and it is not so critical to discuss the AOS mechanism. I can agree that the DC field may not be so important to discuss the mechanism.

We are happy to have convinced referee 1 that the DC field may not be a key ingredient in the explanation of the reversal mechanism.

I would like to give one more chance for the authors to provide more convincing evidence for the in-plane rotation.

1. Please provide the estimation of lattice temperature as a function of the pump fluence. Please provide the measurement of anisotropy energy as a function of temperature. Then, show that the

lattice temperature at the critical fluence matches to the temperature where the anisotropy changes from perpendicular anisotropy to in-plane anisotropy.

The estimation of the lattice temperature has already been given in supplementary material of the manuscript.

Figure of the supplementary material of the submitted paper

Static hysteresis curves have been measured at different temperatures as shown in figure below. A change of the hysteresis loop shape is observed at 150°C, indicative of the in-plane reorientation of the magnetization at this temperature.

Figure: Hysteresis loops measured with out-of-plane field for the sample of composition $Ta(3)/[Tb(0.9)/Co(1.5)]x5/Ta(2)/Pt(3)$ thicknesses in nm.

In order to further reinforce our statement and hopefully convince referee 1, we also did pump probe dynamical hysteresis loops that are reported here below. One can clearly observe that hysteresis loop measured in the polar geometry is characteristic of an out of plane magnetization without any pump. Instead the hysteresis loop measured at 10 or 30ps are characteristic of an in plane magnetization. So we undoubtedly show that at 30ps, the magnetization reorients in plane.

Figure : Hysteresis loops measured by pump-probe method at different delay time with pump fluence of 3.3 mJ/cm^2 for a sample with the composition $\text{TbCo(4)/Co(2)/TbCo(4)}$

2. If a decrease of magnetization signal at $\sim 100 \text{ ps}$ in TRMOKE data is caused by the in-plane rotation, this behaviour will depend on the pump fluence. When the pump fluence is smaller than the critical fluence, the direction of magnetization remains to the out-of-plane direction so that there should be no in-plane rotation. In another word, the decrease of magnetization at $\sim 100 \text{ ps}$ will have a threshold behaviour with the pump fluence. Please demonstrate this point.

Based on the previous measurements of hysteresis loops measured at different time delays, we show clearly that the magnetization reorients in plane. Nevertheless, as requested by the referee 1we report here below the additional measurements.

Figure : Dynamics of the magnetization for different laser fluences with 610 Oe applied field.

Reviewer #2 (Remarks to the Author):

The authors have responded well to all of my concerns. I can now recommend this article for publication.

We are happy to have responded to all concerns of referee 2.

Reviewer #3 (Remarks to the Author):

The authors have mostly responded well to the questions of the reviewers, and made appropriate improvements to the manuscript.

We thank the referee for this positive comment.

However, I am still not convinced that the authors satisfactorily explain why magnetization precession occurs. The authors say when answering Reviewer 1 that "this response is not obvious", and now argue that fluctuations in the anisotropy's orientation is responsible. Further details are missing in their explanation.

We did not change our argumentation between the two versions of the paper. In the first version of the paper, we could not find the origin of the effective field around which the magnetization precesses during its reorientation in plane. Indeed, on the scale of the sample, no anisotropy is well defined in plane. The additional experiment done by MFM showed undoubtedly that the precession has to be considered at a much local scale, the scale of a single grain or of a couple of grains. At this scale, distribution of local anisotropies exists and they are evidenced by the MFM images.

In lines 319-321, the authors write that the direction of anisotropy fluctuates because of the polycrystalline layer. Why is this? Can the authors at least provide a reference, and ideally explain why this occurs?

The referee is right, we have changed "fluctuation" towards "distribution" to make the text more clear. Then we need to consider the following. The bulk anisotropy of a FM layer is linked to its crystallography. For example in the case of Co, the easy axis of magnetization is along (111) in case of Co fcc and (0001) in case of Co hcp. So, without any applied field (including demagnetization field), the magnetization is oriented along those directions. Secondly, by sputtering, when a material is deposited on top of an amorphous layer, which is the case here for FM layer on top of TM-RE alloys layer, several angstroms are needed before building up a crystalline structure and no preferential direction is imposed by the amorphous underlayer (which differs from epitaxy). However, dense crystallographic plans are often favoured parallel to the surface. As a result, we can expect that grains with (111) orientation perpendicular to the surface are favoured, with a large distribution.

How large is this random fluctuation of anisotropy i.e. does it deviate from out-of-plane by a small fraction, or does it vary from in-plane to out-of-plane (over 90degrees)?

We tried to quantify the distribution of the anisotropy axis and also the size of the grains. However, the reduced thickness of the FM layer did not allowed to measure properly a magnetic signal of this layer. All the signals were hidden by the signal of the Pt layer (which is a heavy metal).

How does the laser heating cause the anisotropy to temporarily go in-plane, in order to provide the

necessary torque for precession?

At the beginning of the reversal process, the total magnetization is out of plane. In a thin film, without any anisotropy, the magnetization should be in plane because of the existence of the demagnetization field. Therefore, in our case, the anisotropy of the CoTb is strong enough to keep the magnetization of all the multilayer out of plane. Shining the laser will decrease this anisotropy, the magnetization will temporarily go in plane. This has also been observed by Hennes et al. Phys. Rev. B 102, 174437 (2020). During this reorientation, magnetization will precess along the local direction of anisotropy.

The authors write "the persistence of the magnetization of the TM FM layer due to its high T_c ". Are the authors claiming that the magnetization at the interface (on the RE side) is destroyed, and the magnetization within the FM layer persists? If so, what is the T_c of the FM layer, and how much of its magnetization actually quenches? An estimate should be given.

The referee is right, this paragraph of the paper is slightly confusing and some parts have been clarified. Most of the samples studied (number 3 to 19 in table 1) are made of a (TM-RE alloys / FM) multilayer: the TM-RE alloys layer have high anisotropy and low T_c while the FM layer has a low anisotropy and high T_c (made of Co, Fe, CoFeB or Py). So, after the laser pulse, a fast reduction of the anisotropy and magnetization with the temperature in the TM-RE alloy, related to the very low T_c of the RE, leads to the orientation of the magnetization of the FM layer in plane and to its precession. Indeed, the FM layer is PMA by the exchange interaction with the TM-RE alloy layer. The key ingredient for the reversal is the persistence of the magnetization of the FM layer due to its high T_c (since it is made of a pure Co, Fe, Py, CoFeB layer).

So we are claiming that the magnetization and the anisotropy are strongly reduced in the TM-RE alloy layer while the magnetization within the FM layer persists due to its high T_c . It is difficult to estimate how much the magnetization of FM is quenched.

Linked to this, the authors present interesting new data in Fig. S10, which could actually make a major step towards proving the precessional behavior. In Fig. S10, the magnetic field is changed in strength, and we see a distinct threshold-like behavior. With 377mT and 557mT, the curves are roughly the same (just changed in amplitude), whereas with 196mT, the downward slope becomes much longer (taking up to 200ps to level off). Further description and systematic parameters must be provided here. Were these measurements done with all parameters the same e.g. fluence?

We thank the referee for this comment and for recognizing the added value of the new data presented in Figure S10. Yes, the measurements shown in figure S10 are done for different applied field and a constant fluence of 5.0 mJ/cm². This detailed information has been added to the text.

If the magnetic field is flipped, do we see symmetric behaviour? Can the Kerr signal be actually calibrated, so we can see M_z in terms of 100%, 0% etc (from the imaging, I think this should be straightforward)?

We thank the referee for this comment and suggestion. In the figure below we show the requested measurements. The magnetization has been normalized and the measurements performed for both field directions. We do have symmetric behaviour if we flip the magnetic field (fluence: 5.0 mJ/cm²)

Figure: Dynamics for different applied field and the same fluence 5.0 mJ/cm².

And can the authors provide several more curves with different field, as this would show a clear threshold behavior (consistent with precessional dynamics)?

We do not really understand this point of the referee. As reported in figure S10 as a function of applied field, we clearly see that the oscillations are dependent on the applied field. This new set of data completes, since the first submission of the paper, the large set of additional measurements performed.

Overall, I think the article has benefitted from the review process so far. However, questions still remain. If the authors can provide a suitable response, showing better the precessional dynamics and explaining its origin, the article should be suitable for publication in Nature Communications.

We hope that with these additional information and analysis, the referee 3 will be definitely convinced of the origin of the single pulse reversal in this new class of materials.

Reviewers' Comments:

Reviewer #1:

Remarks to the Author:

The authors revised the manuscript in response to my concern. They showed that perpendicular anisotropy was lost by laser heating in the TRMOKE experiment. This result can support the authors' main claim of the in-plane reorientation. Therefore, I agree that this manuscript can be accepted for publication.

Reviewer #3:

Remarks to the Author:

I have carefully read the revised manuscript and the authors's response letter. Overall, I would now support the publication of the article.

I have a number of comments, hopefully to improve the manuscript further.

1. The authors should include - in the supplementary material - the hysteresis loops measured with the pump-probe experiments (given at top of page 4 in their response letter). This is a good piece of evidence supporting their interpretation. Namely, that the out-of-plane anisotropy dynamically becomes in-plane due to the laser pulse. This data should definitely not be confined to the response letter.

2. An article was published recently on single-shot switching in Tb/Co (<https://doi.org/10.1103/PhysRevResearch.5.023163>), coming to similar conclusions about the switching being precessional. The authors should at least cite this - they may wish to add a sentence or two highlighting the similar/different behavior observed.

3. I would strongly recommend the authors to proof-read their article again - a lot of typos have crept in, with their revised version.